# Discrete Graph Auto-Encoder

**Yoann Boget**                                                                    *yoann.boget@hesge.ch*
*Geneva School for Business administration HES-SO*
*University of Geneva*

**Magda Gregorova**                                                              *magda.gregorova@thws.de*
*Center for Artificial Intelligence and Robotics (CAIRO)*
*Technische Hochschule Würzburg-Schweinfurt (THWS)*

**Alexandros Kalousis**                                                    *alexandros.kalousis@hesge.ch*
*Geneva School for Business administration HES-SO*

**Reviewed on OpenReview:** *https://openreview.net/forum?id=bZ80b0wb9*

## Abstract

Despite advances in generative methods, accurately modeling the distribution of graphs remains challenging primarily because of the absence of a predefined or inherent unique graph representation. Two main strategies have emerged to tackle this issue: 1) restricting the number of possible representations by sorting the nodes, or 2) using permutation-invariant/equivariant functions, specifically Graph Neural Networks (GNNs).

In this paper, we introduce a new framework named Discrete Graph Auto-Encoder (DGAE), which leverages both strategies' strengths and mitigates their respective limitations. In essence, we propose a strategy in two steps. We first use a permutation-equivariant auto-encoder to convert graphs into sets of discrete latent node representations, each node being represented by a sequence of quantized vectors. In the second step, we sort the sets of discrete latent representations and learn their distribution with a specifically designed auto-regressive model based on the Transformer architecture.

Through multiple experimental evaluations, we demonstrate the competitive performances of our model in comparison to the existing state-of-the-art across various datasets. Various ablation studies support the interest of our method.

## 1 Introduction

Graph generative models are a significant research area with broad potential applications. While most existing models focus on generating molecules for *de novo* drug design, there has been interest in using these models for tasks such as material design (Lu et al., 2020), protein design (Ingraham et al., 2019), code programming modeling (Brockschmidt et al., 2019), semantic graph modeling in natural language processing (Chen et al., 2018; Klawonn & Heim, 2018), or scene graph modeling in robotics (Li et al., 2017).

Despite advances in generative methods, accurately modeling the distribution of graphs remains a complex task. The main challenge lies in the absence of predefined or inherent unique graph representations. Given a graph of $n$ nodes, there are $n!$ permutations, each corresponding to a distinct representation of the same graph.

Two main strategies have emerged to tackle this issue:

1. The first accepts multiple representations of the same graph but restricts their quantity by sorting the nodes thanks to a heuristic such as Breadth-First Search (BFS), as in (You et al., 2018; Shi et al., 2020; Luo et al., 2021).

2. The alternative approach employs models that are inherently invariant to node ordering permutations. This method achieves permutation invariance through the use of Graph Neural Networks, as in (Vignac et al., 2023; Jo et al., 2022).

Both methods have their own challenges, which we elaborate on in Section 2.

Our main contribution is introducing a new generative model designed to address this issue of multiple graph representations. We use a permutation-equivariant auto-encoder to convert graphs into sets of node embeddings and reconstruct graphs from sets of node embeddings. Contrary to graphs, any sorting algorithm yields a unique representation for sets. So, we transform the sets into sequences and leverage an auto-regressive model to learn the implicit distribution over sets.

In implementing this framework, we introduce new methods and techniques or adapt existing ones. We enhance the node embeddings by introducing an innovative feature augmentation strategy. To ease the modeling of the latent distribution, we remove unnecessary information and reduce the size of the latent space by quantizing the node representations. However, for pragmatic considerations, we initially partition the node embeddings and subsequently quantize these partitions. As a result of this sequentialization, the latent representations form sequences spanning two dimensions: the partitions and the nodes. To model this distribution, we introduce a novel Transformer architecture designed to take advantage of its 2-dimensional structure.

The development and subsequent evaluation of our model yield several significant contributions that we summarize as follows:

- We introduce a novel generative model for graphs, leveraging a graph-to-set discrete auto-encoder invariant to permutations.
- We develop a new Transformer architecture, the 2D-Transformer, specifically designed to model sequences across two dimensions.
- We design a new technique for graph feature augmentation, which we call $p-$path features.
- For the first time, we empirically evaluate various feature augmentation strategies via an ablation study.
- Through empirical analysis, we demonstrate the competitive performances of our model in comparison to the existing state-of-the-art across various datasets.

In the following, Section 2 presents the foundational concepts and issues of graph modeling. Section 3 discusses the literature on graph generation. In Section 4, we introduce our Discrete Graph Auto-Encoder. Lastly, Section 5 presents the experimental evaluation of our model, demonstrating its competitive performance against baseline models for various datasets, including both simple and annotated graphs.

## 2 Background

In this section, we provide the necessary background about graph modeling. We define some notations and then review the challenges of learning graph representations, which mainly stems from the graph isomorphism problem. Finally, we introduce Graph Neural Networks (GNN) and discuss some of their limitations.

### 2.1 Notation

We define a simple (unannotated and undirected) graph as a tuple $\mathcal{G} = (\mathcal{V}, \mathcal{E})$, where $\mathcal{V}$ is the set of nodes, whose cardinality is $n$, and $\mathcal{E}$ is the set of edges between these nodes. Given two nodes $\nu_i$ and $\nu_j$ in $\mathcal{V}$, we represent the edge connecting them as $\epsilon_{i,j} = (\nu_i, \nu_j) \in \mathcal{E}$. For annotated graphs, we introduce two additional functions $V$ and $E$ that map the nodes and the edges to their respective attributes: $V(\nu_i) = \boldsymbol{x}_i \in \mathbb{R}^R$ and $E(\epsilon_{i,j}) = \boldsymbol{e}_{i,j} \in \mathbb{R}^S$. In the case of univariate discrete attributes, the categorical attributes are one-hot encoded, and $R$ and $S$ are the number of node and edge categories, respectively. We denote an annotated graph as $\mathcal{G} = (\mathcal{V}, \mathcal{E}, V, E)$.

## 2.2 Graph isomorphism

A fundamental property of graphs is their invariance with respect to the representation ordering of the node. To put it differently, any permutation applied to the representation ordering of the node will yield an identical graph. Therefore, there are $n!$ possible isomorphic representations of the same graph. In the following, we employ the notation $\pi(\cdot)$ to indicate that we represent the graph under a specific node ordering.

The graph isomorphism problem, which consists of determining whether two graphs are identical or distinct, follows from these possible multiple representations (Köbler et al., 1993). Algorithms that offer a unique representation for each isomorphism are at least as expensive computationally as solving the graph isomorphism problem[1]. This problem has not been proven to be solvable in polynomial time (Helfgott et al., 2017).

As a result, algorithms that uniquely sort nodes in any graph are, at least, as expensive as algorithms solving the graph isomorphism problem.

Graph isomorphism is a significant issue for graph generative models. Without a specific solution, generative models must learn a new set of parameters for each permutation. Given the vast number of possible permutations for each graph, this brute-force approach is unreasonable. In fact, to the best of our knowledge, no generative model for graphs follows such a procedure.

## 2.3 Generative models and graph isomorphism

Previous generative models, detailed in Section 3, have predominantly pursued two strategies to deal with the issue of ordering: sequentialization and invariance to permutation.

### 2.3.1 Sequentialization

The first method aims to establish an ordering of nodes that yields a sequentialization of the graph. Most models adopt algorithms sorting nodes in a non-unique manner. Breadth-first search (BFS), used for instance in (You et al., 2018; Shi et al., 2020; Luo et al., 2021), is the most common heuristic to order nodes in graph generative models. We presume that BFS is also a convenient way of traversing the graph and works as an inductive bias for auto-regressive models. The main drawback of BFS is that the number of potential representations can still be considerable. In the worst case (star graph), BFS does not reduce the number of possible representations at all.

### 2.3.2 Invariance to permutation

The alternative approach circumvents the issue by developing models invariant to permutation by construction. These models rely on permutation-invariant and permutation-equivariant functions. Permutation-invariance is defined as:

$$f(\mathcal{G}) = x \iff f(\pi(\mathcal{G})) = x. \tag{1}$$

A function $f$ is said permutation-equivariant if:

$$f(\pi(\mathcal{G})) = \pi(f(\mathcal{G})). \tag{2}$$

Notably, equivariance to permutation implies that input and output have the same cardinality. In practice, Graph Neural Networks (GNN) entails permutation-invariance and permutation-equivariance. In the next Section (2.4), we introduce the fundamental principles of GNNs

## 2.4 Graph neural networks

GNN layers are permutation-equivariant functions. Since the cardinality of the input and the output of such function must match, GNNs are said to be 'flat'[2].

---

[1]The graph isomorphism problem involves determining whether two graphs are isomorphic. Establishing a unique representation for each isomorphism can be considered a way to solve the problem. If the representations match, then the graphs are isomorphic.

[2]It does not mean that the dimension of each node representation, usually a vector, does not change.

An aggregating function providing graph-level information often follows the GNNs layers. Such an aggregation with a commutative function, as summation or averaging, results in a permutation-invariant function.

### 2.4.1 Message Passing Neural Networks

Message Passing Neural Networks (MPNNs) (Scarselli et al., 2009) represent a prevalent class of Graph Neural Networks (GNNs), even though other classes of GNN have gained in popularity, in particular, GNN, where each node shares information with all the other nodes as, for instance, in (Yun et al., 2019). The core concept underlying MPNNs involves updating the representation of each node by aggregating information from its neighboring nodes, those with which it shares an edge. Consequently, after $l$ layers, each node aggregates information from its $l$-hop neighborhood. We refer to this $l$-hop neighborhood as the receptive field of node features. So, each layer increases the size of the receptive field.

**Message Passing Layer**  The update of the $l^{\text{th}}$ layer of an MPNN involves two steps. The first step computes the message:

$$e_{i,j}^{l+1} = f_{\text{edge}}^l(x_i^l, x_j^l, e_{ij}^l) \tag{3}$$

The second step aggregates information to compute a new node representation:

$$x_i^{l+1} = f_{\text{node}}^l\left(x_i^l, \bigoplus_{j \in \mathcal{N}(i)} e_{i,j}^{l+1}\right) \tag{4}$$

In these equations, the functions $f$ are small neural networks, the $\bigoplus$ denotes any commutative function, and $\mathcal{N}(i)$ is the set of nodes connected to node $i$. Note that the 'message' $e_{i,j}^{l+1}$ can also be interpreted as an edge representation and passed to the next layer. In the following, we will favor this interpretation.

**Limitations**  MPNNs present the advantage of addressing the graph isomorphism problem but exhibit some shortcomings. Specifically, two challenges have been widely acknowledged: oversmoothing and limitation of their expressive power.

Firstly, all node and edge features tend to the same value as the number of layers increases. We call this phenomenon oversmoothing (Li et al., 2018-02; Oono & Suzuki, 2020). Because of it, one should limit the depth of MPNNs to a few layers.

Secondly, MPNNs exhibit limited expressive capacities in the sense that they cannot discern specific pairs of non-isomorphic graphs (Morris et al., 2019). Standard MPNNs are, at best, as expressive as the Weisfeiler-Lehman (1-WL) test (Xu et al., 2019; Loukas, 2019). Consequently, they fail to detect some basic substructures (Arvind et al., 2020; Chen et al., 2020). We illustrate such failure in Appendix D.

Two strategies to enhance the expressive capability of standard MPNNs have been proposed. Developing GNNs more powerful than MPNNs is the first strategy (Maron et al., 2019; Vignac et al., 2020). So far, such GNNs are computationally much more expensive and, therefore, ineffective.

The second approach involves the augmentation of the node features with synthetic attributes, including spectral embeddings, cycle counts, and the integration of random noise. The feature augmentation enhances the ability of GNNs to capture the graph structure. Using spectral embeddings (Laplacian Positional Encoding), cycle count, or random noise is now usual as additional synthetic features for graph generation (Krawczuk et al., 2021; Vignac et al., 2023). We present the details about these features in appendix D. In Section 4.1.2, we propose a new method for feature augmentation called $p$-path features, and in Section 5.4, we assess experimentally the effect of the various methods on our model.

# 3 Related work

In this Section, we briefly review existing work on graph generative models. As exposed in Section 2, generative models for graphs have followed two main strategies to address the graph representation issue: sequentialization and invariance to permutation.

## 3.1 Sequential generation

This sequentialization serves dual purposes: it restricts the number of possible graph representations and facilitates auto-regressive graph generation. Notably, all models following this approach are auto-regressive. Graph canonization is computationally expensive, as discussed in Section 2. To the best of our knowledge, GraphGen (Goyal et al., 2020) stands as the only model employing this particular strategy for generic graphs. Most works only seek to reduce the number of graph representations by adopting a Breadth-First Search (BFS) based method.

We can further categorize auto-regressive models into two classes. The first class, which includes the seminal GraphRNN (You et al., 2018), employs a recurrent framework that necessitates maintaining and updating a global graph-level hidden state. This approach inherently introduces long-range dependencies, as proximate nodes in graph topology can be distant in the sequential representation. The second class of auto-regressive models sidesteps long-range dependencies by recomputing the state for the partially generated graph at each step (Shi et al., 2020; Luo et al., 2021; Liao et al., 2019; Kong et al., 2023). Although this alleviates the long-range dependency issues, it drastically increases the computational cost.

Furthermore, auto-regressive models, generating nodes and edges one by one, are inherently slow during the generation phase. Generation slowness is especially true for the second class of models, as highlighted by experimental results in Section 5.3.2 and in (Jo et al., 2022). GRANs (Liao et al., 2019) attempt to address this speed issue by generating multiple nodes and edges simultaneously, albeit with a trade-off in generation performance.

Models previously discussed are generic, meaning they operate independently of any specific domain knowledge. However, many models are tailored to particular domains, notably in the domain of molecule generation, which represents a predominant application of graph generative modeling. Such domain-specific models often employ canonical representations. The Canonical SMILES notation, for instance, is commonly used to represent molecules, as in (Gómez-Bombarelli et al., 2018; Kusner et al., 2017). Recently, powerful models representing molecules as 3D objects have emerged (Hoogeboom et al., 2022; Xu et al., 2023). While plenty of domain-specific sequential models exist (Liu et al., 2018; Samanta et al., 2020; Jin et al., 2018; 2020; Kuznetsov & Polykovskiy, 2021; Li et al., 2018), in this work, we focus on developing a generic graph model.

## 3.2 Models invariant to permutations

The second main strategy to address the multiple potential representations of a graph consists of using models invariant to permutations. By definition, these models, also referred to as 'one-shot', cannot be sequential. These models implement invariance or equivariance through GNNs. They are thus insensitive to the ordering of the graphs during training. Various methods, such as Generative Adversarial Networks (GANs), Normalizing Flows (NF), and diffusion/score-base models, have been proposed following this strategy, each with its merits and issues.

In GANs, a discriminator (or critic) classifies (or scores) graphs between true (from the dataset) and generated, while a generator aims to maximize the error of the discriminator. MolGAN (De Cao & Kipf, 2018) proposes to use permutation-invariant discriminator. GG-GAN (Krawczuk et al., 2021) add a permutation-equivariant generator, and GrannGan (Boget et al., 2023) proposes to generate only the nodes and edges attribute from sampled skeletons. However, the adversarial training is known to be unstable. MolGAN exhibits mode collapse and low diversity. GG-GAN also suffers from a lack of diversity, while GrannGAN does not generate graphs, but only their attributes.

Normalizing Flows are designed to learn an invertible mapping between the data distribution and a known continuous distribution, typically the Standard Normal distribution. Both GraphNVP (Madhawa et al., 2019) and Moflow (Zang & Wang, 2020) use a two-step process, first generating an adjacency tensor, followed by an annotation matrix conditioned on this tensor. GNF (Liu et al., 2019) needs only one step but focuses on generating simple graphs only.

However, we hypothesize that, even with dequantization, efficiently spanning the continuous latent space using discrete objects remains a significant challenge. Indeed, based on their empirical evaluations, while these models surpass the performance of GAN-based approaches, they do not model graph distributions as effectively as auto-regressive models and those relying on diffusion/score-based methods.

Diffusion and score-based models gradually introduce Gaussian noise to the data and learn to denoise it for each noise level. In graphs, some models have suggested adding Gaussian noise to the adjacency matrix (Yang et al., 2019) and to the annotation matrix (Jo et al., 2022). By doing so, these models generate fully connected graphs, facilitating direct information sharing among all nodes.

However, as stated in (Vignac et al., 2023), the continuous noise compromises the graph structures, leading them to collapse into fully connected graphs. To address this concern, DiGress introduced a discrete diffusion model for graphs (Vignac et al., 2023), which integrates discrete noise to maintain a coherent graph structure at every noise level. In practice, however, the discretization does not appear to bring significant improvement over GDSS.

Whether continuous or discrete, the diffusion/score-based approach effectively captures the global structure of the graph and, as of now, has produced the best results with Digress and GDSS (Jo et al., 2022) considered as state-of-the-art. However, the extensive number of denoising steps required by these models make the generation process comparatively slow, as shown experimentally and reported in Section 5.

### 3.3 Graph Auto-Encoders

Closer to our work, Kipf and Welling (Kipf & Welling, 2016) proposed a permutation-equivariant graph auto-encoder, namely the Graph Auto-Encoder (GAE) and the Variational Graph Auto-Encoder (VGAE) relatively long ago. However, they use it as a feature extractor for downstream tasks rather than for generation. In fact, neither of these two models captures the interaction and dependencies between node embeddings. One of our main contributions consists of modeling these interactions and dependencies necessary for generation. On a side note, we also remark that they use a simple activated scalar product as a decoder, which can neither capture interactions between multiple node embeddings nor decode node and edge attributes. Like most auto-encoders in generative modeling, we use a decoder mirroring the encoder.

Lastly, GraphVAE (Simonovsky & Komodakis, 2018) adopts a distinct approach to handle the multiple possible graph representations. A permutation-invariant encoder captures a graph-level latent representation, losing the original node ordering in their aggregation. Then, a decoder, built using a standard feed-forward neural network, aims to reconstruct the original graph to yield a graph in any permutation. Nevertheless, during the training phase, the model requires aligning the original representation with its reconstructed counterpart. This comparison necessitates an expensive graph-matching procedure with a computational complexity of order $\mathcal{O}(n^4)$.

Unlike GraphVAE, our auto-encoder is permutation-equivariant end-to-end. It does not to capture the graph representation in a single vector but a in a set of node embeddings, each capturing its local neighborhood so that we can reconstruct the graph from its set of node embeddings. Since the sequentialization of sets does not suffer the same problem as graphs, we can model the set of node embeddings as a sequence leveraging an auto-regressive model. To sum up, we follow the permutation invariance strategy to train a graph-to-set auto-encoder, and then we follow the sequentialization strategy to model the distribution over sets.

## 4 Discrete Graph To Set Auto-Encoder

In a nutshell, our approach involves a two-step strategy. The first step is a graph auto-encoder. We encode a graph into a set of $n$ node embeddings. For practical reasons detailed hereunder, we partition each node

embedding into $C$ vectors and quantize each vector independently. Then, we decode the quantized latent representation, aiming to recover the original graph. This first step generates a discrete latent distribution over sets of node embeddings, yet this latent distribution remains unknown.

In the second step, our objective is to learn this latent distribution. Rather than directly modeling a distribution over sets, we transform the sets into sequences. We then leverage a model based on the Transformer architecture to learn these sequences auto-regressively.

The rationale behind this two-step strategy stems from using permutation-equivariant functions, particularly Message Passing Neural Networks (MPNNs), to project the graph into a latent space and decode it. This approach prevents the node ordering issue, making the computation of the reconstruction loss straightforward without needing a matching procedure. However, as discussed in Section 2.4.1, MPNNs present inherent limitations, such as being constrained in depth and only capturing information from their $L$-hop neighborhood, $L$ being the number of layers in the MPNN. Consequently, MPNNs are unable to model interactions between distant nodes. Therefore, we introduce the second step to model the latent distribution to address these long-range dependencies between node embeddings.

In this section, we present and motivate these two steps in detail (Sections 4.1 and 4.2), and we explicitly explain how we perform generation (Section 4.3. The source code of our model is publicly available at `https://github.com/yoboget/dgae`.

## 4.1 Discrete Graph Auto-Encoder

In this initial step, our primary objectives when encoding graphs are twofold. Firstly, we aim to learn node embeddings that facilitate the accurate reconstruction of the original graph. Secondly, we strive to create a latent distribution that is handy to learn in the second step.

To achieve this, we enhance the input graph representation with feature augmentation (Section 4.1.1). We use a Message Passing Neural Network (MPNN) to encode graphs $\mathcal{G}$ into sets of node embeddings $\{z_i^h, ..., z_n^h\} = \mathcal{Z}^h$, mapping a space over graphs to a space over sets $f_{encoder} : \mathbb{G} \mapsto \mathbb{Z}^h$ (Section 4.1.2). Partitioning and quantizing the node embeddings results in a handy discrete latent space with known support, facilitating the modeling of the latent distribution in the second step. Specifically, we partition the node embeddings in a sequence of $C$ vectors $z_i^h = (z_1^h, ..., z_C^h)$. We then quantize each vector independently (Section 4.1.3), replacing them with their closest neighbor in a codebook. Finally, another MPNN decodes the sets of quantized vectors $\mathcal{Z}^q$, also called codewords, to graphs $\hat{\mathcal{G}}$, formally $f_{decoder} : \mathbb{Z}^q \mapsto \mathbb{G}$. Figure 1 visually summarizes the entire procedure.

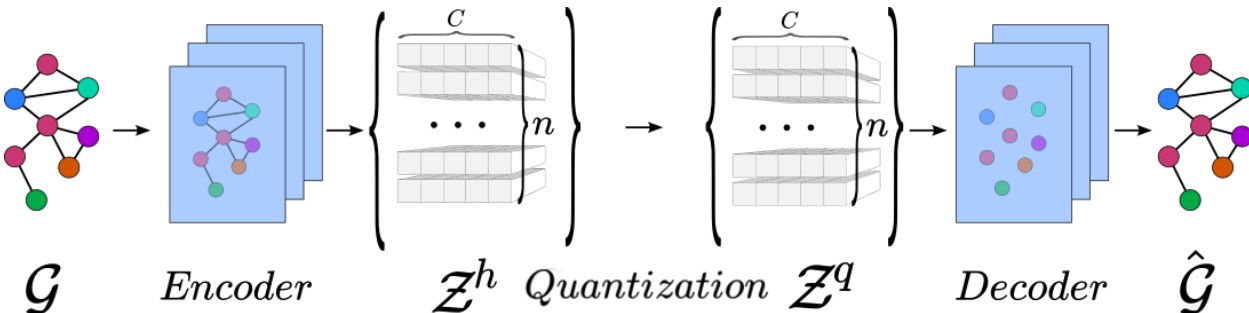

Figure 1: Diagram of our auto-encoder. 1. The encoder is an MPNN transforming the graph into a set of node embeddings $\mathcal{Z}^h$. 2. The elements of the set $\mathcal{Z}^h$ are partitioned and quantized, producing a set of codeword sequences $\mathcal{Z}^q$. 3. The decoder, an other MPNN, takes the set $\mathcal{Z}^q$ and reconstruct the original graph.

### 4.1.1 Input Graph and Feature Augmentation

As explained in Section 2.4, conventional MPNNs cannot identify certain fundamental graph substructures. Such a shortcoming adversely impacts the information captured by the encoder. Inevitably, the node em-

beddings miss the information the encoder fails to capture, preventing accurate reconstruction. A common mitigation strategy involves enhancing MPNN capacities by augmenting the input features with synthetic attributes. We employ this approach to generate more informative node embeddings, allowing for better reconstruction performance.

Here, we propose a new augmentation scheme that aggregates information from the $p$-path neighborhood. Specifically, we first create virtual edges (with a vector of zeros as attributes) between nodes that are connected by paths of length 2 to $p$ (the edges between adjacent nodes, i.e. connected by a path of length 1, are already represented by a vector of edge attributes). We then concatenate a synthetic $p-$dimensional vector to each vector of edge attribute, whose $i^{th}$ entry corresponds to the number of paths of length $i$ connecting the two endpoints. Similarly, we augment the node features with a $p-$dimensional vector, whose $i^{th}$ entry indicates the number of paths of length $i$ emanating from the node. In Appendix D, we provide more details on how we compute the path lengths as well as information about the other feature augmentation schemes used for our experiments.

Our method presents two benefits compared to other methods. First, unlike other methods, our $p-$path augmentation strategy produces synthetic attributes for nodes *and* edges, providing richer information about the substructures around each node. Second, the virtual edges aggregate information from a larger neighborhood. The computation of the synthetic features is a unique prepossessing step. In our experiments, we use the $p-$path method setting $p = 3$.

Our solution is related to the recent $k-$hop message passing presented in (Feng et al., 2022), both using the path length information. Our proposition differs on two main points. First, we use the path information to create synthetic features, while $k-$hop message passing uses this information to compute different message passing functions for each path length. Second, we take into account the number of paths of length $p$ between 2 nodes, while they only consider if there is at least one such path.

In Section 5.4, we empirically show that our method increases the reconstruction performance of our auto-encoder the most compared to other methods.

### 4.1.2 Encoder

We aim to encode graphs as sets of node embeddings, where each embedding captures its local graph structure so that we can effectively recover the original graph from these embeddings. We use an MPNN as encoder since they are specifically designed and efficient for modeling the local neighborhood of each node.

We formally define our encoder as an $L$-layer MPNN whose layers follow these two equations:

$$e_{i,j}^{l+1} = \text{bn}(f_{\text{edge}}([\boldsymbol{x}_i^l, \boldsymbol{x}_j^l, \boldsymbol{e}_{i,j}^l])) \tag{5}$$

$$\boldsymbol{x}_i^{l+1} = \text{bn}\left(\boldsymbol{x}_i^l + \sum_{j \in \mathcal{N}(i)} f_{\text{node}}([\boldsymbol{x}_i^l, \boldsymbol{x}_j^l, \boldsymbol{e}_{i,j}^l])\right), \tag{6}$$

where bn stands for *batch normalization* (Ioffe & Szegedy, 2015) and $[\cdot, \cdot]$ is the concatenation operator. After the last layer, we discard the edge representation $\boldsymbol{e}_{i,j}^L$ and keep only the node representation so that $\boldsymbol{z}_i^h = \boldsymbol{x}_i^L \in \mathbb{R}^{d_{latent}}$. The complete graph latent representation is the set of all the node embedding $\mathcal{Z}^h = \{\boldsymbol{z}_1^h, ..., \boldsymbol{z}_n^h\}$ where the superscript $h$ indicates the set before quantization. The superscript $q$ will refer to the latent representations after quantization.

### 4.1.3 Discrete Latent Distributions

In the context of our auto-encoder, quantization consists of replacing a $d$ dimensional vector by its nearest neighbor in a learned codebook $H \in \mathbb{R}^{m \times d}$, $m$ being the codebook size. By partitioning, we refer to a simple reshaping operation, where a vector of dimension $d$ is reshaped in $C$ vectors of dimension $d/C$. Both

operations follow the same objectives: produce a latent distribution that is handy to model (in the second step) and flexible enough to yield good reconstruction.

Specifically, the encoder outputs $n$ node embeddings $\boldsymbol{z}_i^h \in \mathbb{R}^{d_{latent}}$. So, we partition these node embeddings in $C$ vectors $(\boldsymbol{z}_{i,1}^h, ..., \boldsymbol{z}_{i,C}^h)$, where each $\boldsymbol{z}_{i,c}^h$ is in $\mathbb{R}^{d_{latent}/C}$. Then, we quantize each vector independently. We refer to the partitioned and quantized representations as the discrete latent representation $\mathcal{Z}^q = \{(\boldsymbol{z}_{1,1}^q, ..., \boldsymbol{z}_{1,C}^q), ..., (\boldsymbol{z}_{n,1}^q, ..., \boldsymbol{z}_{n,C}^q)\}$. Figure 2 depicts the quantization procedure. Partitioning and quantization result in discrete latent distributions.

Figure 2: Diagram of the quantization. We represent each node embedding by $C$ partition vectors $\boldsymbol{z}_{i,c}^h$. Then, we quantize each of these vectors by replacing them with their closest neighbor from the corresponding codebook $H_c$. The vectors in the codebooks are parameters learned during training.

**Discrete Latent Representations** The node embeddings $\boldsymbol{z}_i^h$, as they are output by the encoder, are real-valued vectors in $\mathbb{R}^{d_{latent}}$. We make two observations regarding these vectors.

First, if the original graphs have only discrete node and edge attributes, the support of this distribution is discrete. It corresponds to all the possible subgraphs within the $L$-hop neighborhood, where $L$ is the number of encoder layers. However, enumerating the support of this discrete distribution is intractable in almost all practical scenarios, which makes the modeling of this discrete distribution unpractical. In the continuous case, the distribution of the node embeddings remains also unknown and challenging to model.

Second, the encoder cannot model interactions between distant nodes. By consequence, even if we could model the distribution of each node embedding individually, the distribution over the sets of node embeddings would remain unknown.

Both observations, i.e., the potential discrete support of the node embedding distributions and the impossible modeling of distant nodes, make standard variational auto-encoder unappropriated for the task. More generally, the distribution of node embeddings $\boldsymbol{z}_i^h$, as they are output by the encoder, is challenging. On the other hand, discrete latent models have shown impressive results with both discrete and continuous data (van den Oord et al., 2017; Razavi et al., 2019). Therefore, we leverage such a model using quantization to produce discrete latent distributions with known support.

**Quantization** The quantization of the node embeddings into discrete latent representations presents many advantages. First, discrete latent representation is a natural fit for modeling discrete structures like graphs. In particular, discrete representation prevents approximating step functions or Dirac delta functions with neural networks, which require smooth differentiable functions.

Quantization also enforces the codewords to follow a $m-$dimensional categorical distribution, where $m$, the codebook size, is a hyper-parameter. Knowing that the latent vectors follow an $m-$dimensional categorical distribution, we can learn their parameters in the second stage of our model.

Moreover, quantization acts as an information bottleneck. Assuming no prior information, the information contained in each selected codeword is $I = log_2(m)$ bits and depends only on the codebook size $m$. By training the auto-encoder with a reconstruction objective, we encourage the quantization process to retain only the most valuable information for reconstruction. Also, by reducing the codebook size, we shrink the number of categories and, therefore, the number of parameters of the categorical distribution, making the latent distribution easier to learn. Limiting the codebook size also reduces the risk of codebook collapse (Kaiser et al., 2018) as observed in our experiments (See Section 5.4), favoring a better coverage of the space and so preventing sampling from regions out of the distribution. However, restraining the information passed to the decoder makes the reconstruction harder. As a result, there is a trade-off between the reconstruction quality and the dimensionality of the categorical distribution stemming from the codebook size. We empirically show the effect of this trade-off in our experiments (See Section 5.4). We indeed observe that the reconstruction loss decreases when the codebook size increases. However, larger codebook sizes do not yield better generation performances despite better reconstruction performance. We explain this observation because larger codebooks result in larger categorical distributions that are harder to learn.

**Partitioning**   In practice, we often need a large discrete latent space to model each node representation with enough flexibility. We can obtain this flexibility either with a single codeword with a large codebook or with multiple codewords, each coming from a (much) smaller codebook.

Large codebooks imply high dimensional categorical distributions, which are inconvenient in practice. Specifically, modeling a high-dimensional categorical distribution in the second stage of our model would require an output layer whose size corresponds to the dimension of this distribution. Large output layers, particularly those larger than their hidden layers, are ineffective in practice. Our experiments show that our model's generation performance drops for the larger categorical distribution.

So, instead of representing each node embedding with a single codeword, we use multiple codewords for each node representation. To that end, we partition the node embeddings $\boldsymbol{z}_i^h \in R^{d_{latent}}$ in $C$ vectors $\boldsymbol{z}_i^h, c \in R^{d_{latent}/C}$ by reshaping them.

In consequence, each node embedding is represented by $C$ partition vectors. We quantize each of these partition vectors independently, each partition with its own codebook. In fact, in graphs, as in images, codebooks shared across features reflect an invariance assumption: the permutation invariance for graphs and the translation invariance for images. We use different codebooks for the different partitions as we do not assume any invariance across partitions. In other words, there is no reason to share codebooks across partitions because the various partitions do not belong to the same feature map.

After partitioning and quantization, the latent representation of each node is a sequence of codewords. We call dictionary the set all possible codeword arrangements. Therefore, for a fixed codebook size $m$, the dictionary size is $M = m^C$. We empirically assess the effect of partitioning and present the results in Section 5.4.

**Formalisation**   Specifically, the partition function $f_p$ defines a simple reshaping operation transforming a vector of dimension $d_{latent}$ in $C$ vectors of dimensions $d_{latent}/C$, that is $f_p : \mathbb{R}^{d_{latent}} \mapsto \mathbb{R}^{C \times (d_{latent}/C)}$.

We define the quantization function $q_c : \mathbb{R}^{d_{latent}/C} \mapsto \{\boldsymbol{h}_{1,c}, ...\boldsymbol{h}_{m,c}\}$ as the mapping of the $c$-th partition of node $i$, to its closest neighbor in the corresponding codebooks $H_c \in \mathbb{R}^{m \times d_{latent}/C}$, such that:

$$q_c(\boldsymbol{z}_{i,c}^h) = \boldsymbol{z}_{i,c}^q = \boldsymbol{h}_{k,c} \quad \text{with} \quad k = \arg\min_g(||\boldsymbol{z}_{i,c}^h - \boldsymbol{h}_{g,c}||_2). \tag{7}$$

Notably, the quantization procedure acts independently on every element in the set. It is, therefore, permutation-equivariant.

We underline that a bijection exists between the quantized vectors and their corresponding indices in the codebooks. Therefore, the quantized vectors and their indices hold the same information. In the following, we refer to the set of indices corresponding to the set of quantized vectors as $\mathcal{Z}^q = \{(\boldsymbol{z}_{1,1}^q, ..., \boldsymbol{z}_{1,C}^q), ..., (\boldsymbol{z}_{n,1}^q, ..., \boldsymbol{z}_{n,C}^q))$ as $\mathcal{K} = \{(k_{1,1}, ..., k_{1,C}), ..., (k_{n,1}, ..., k_{n,C})), \text{ where } k_{i,c} \in \{1, ..., m\}$.

### 4.1.4   Decoder

After quantization, the decoder has to recover the graph structure and the node and edge attributes for annotated graphs. Formally, it is a set-to-graph function $f_{decoder} : \mathbb{Z}^q \mapsto \mathbb{G}$. Since we have dropped any explicit representation of the graph (see Section 4.1.2), when we decode it, we assume a fully connected graph among the elements of $\mathcal{Z}^q$. We subsequently feed the fully connected graph into an MPNN similar to the one used as encoder but without feature augmentation. The decoder output serves as logit to model the discrete distributions over the edges and the nodes when the graph has node attributes. We present the implementation details in Appendix A.1.

### 4.1.5   Training

Our training strategy follows principles developed to train models performing quantization of the latent space (van den Oord et al., 2017). The training objectives are simultaneously reconstruction and quantization. We define a reconstruction loss for the first objective. We update the codebook vectors center using Exponential Moving Average (EMA) as the second objective. Lastly, we add a regularization term to prevent the latent space from growing indefinitely. In the following, we present these elements and provide the implementation details in Appendix A.2.

**Reconstruction loss**   We train the parameters of MPNNs (encoder and decoder) to minimize the reconstruction loss. We define it as the expected negative log-likelihood of the graph given the set of codewords $\mathcal{Z}^q$:

$$\mathcal{L}_{recon.} = -\mathbb{E}_{\mathcal{Z}^q} \ln \left( p_\theta(G|\mathcal{Z}^q) \right) \tag{8}$$

Thanks to the auto-encoder's permutation-equivariance, the loss computation is straightforward since the representation order of the output corresponds to the input one.

The main difficulty comes from the quantization operation, which is non-differentiable. Therefore, no gradient can flow back to the encoder. We circumvent this difficulty by following (van den Oord et al., 2017) and passing the gradient with respect to the codeword directly to the encoder's output, bypassing the quantization operation.

**Vector Quantization Objective**   The Vector Quantization (VQ) objective updates the vector in the codebooks. The objective function, acting as an online $k$-means algorithm, updates the vectors in the codebook using EMA as proposed in (van den Oord et al., 2017). We adopt the k-mean++ initialization for the codebooks as proposed in (Łańcucki et al., 2020).

**Commitment loss**   Finally, nothing prevents the embeddings from taking arbitrarily large values. To circumvent this issue, we follow again (van den Oord et al., 2017). We incorporate a commitment loss function, which pushes the encoder outputs close to their corresponding codewords.

## 4.2   Modeling The Latent Distribution

The auto-encoder implicitly defines a distribution over the latent space $\mathbb{Z}^q$, which translates over the space of the corresponding sets of indices $\mathbb{K}$ defined in Section 4.1.3. For generation, we need to sample from one of these distributions. So, we decide to model the distribution of indices. After sampling, we recover the corresponding codewords from their indices.

Instead of learning a distribution over the sets of index partitions $\{(k_{1,1}, ..., k_{1,C}), ..., (k_{n,1}, ..., k_{n,C})\}$, we turn the sets into sequences by sorting them. After sorting, the sequences of indices $\mathrm{K} \in \mathbb{R}^{n \times C}$ have two dimensions: the partition dimension and the node dimension. Instead of flattening the sequence, we propose a new model based on the Transformer architecture (Vaswani et al., 2017) that leverages the 2-dimensional structure of the sequences to implement parameter sharing for computational efficiency and better generalization. We call it 2D-Transformer.

Our main contribution concerns the attention function. Before presenting it, we first detail the sequentialization, the auto-regressive process, the input and output of our model, and recall some basics of the Transformer architecture.

### 4.2.1 Sequentialization

The existence of multiple permutations carries over from the graphs to the sets in the latent space. The multiple possible representations as well as the limitations of permutation-equivariant and permutation-invariant functions are still an issue.

However, unlike graphs, sorting sets in a unique representation is easy. Any sorting algorithm can sort sets, yielding a unique representation. We can then represent the latent distributions as sequences and learn them auto-regressively.

Additionally, we remark that sorting is a specific kind of permutation. Since the decoder is permutation-equivariant, permuting the node embeddings in the latent space only changes the representation of the output graph but not the graph itself. So, we sequentialize the sets and learn the sequential distribution auto-regressively.

In practice, we sort the set of index sequences $\mathcal{K} = \{(k_{1,1}, ..., k_{1,C}), ..., (k_{n,1}, ..., k_{n,C})\}$ (the indices corresponding to the codewords $\mathcal{Z}^q = \{(\boldsymbol{z}_{1,1}^q, ..., \boldsymbol{z}_{1,C}^q), ..., (\boldsymbol{z}_{n,1}^q, ..., \boldsymbol{z}_{n,C}^q)\}$, see Section 4.1.3) by increasing order using the first partition as the primary criterion. In case of a tie, we use the index of the second partition as a secondary criterion, etc. We permute the set $\mathcal{Z}$ accordingly. We obtain sequences of indices $\mathrm{K} \in \mathbb{R}^{n \times C}$ and, the corresponding sequence of codewords $Z \in \mathbb{R}^{n \times C \times d_{latent}/C}$.

Since we perform the sorting operation as a preprocessing step for the second stage of our model, there is no backpropagation through the sorting operation.

The ordering obtained is arbitrary, but it has the advantage of being completely generic. Domain-specific information can certainly lead to orderings that improve learning performance. We let this question open for further domain-specific research.

Regarding notation, we remove the superscript $q$ from the latent vector $\boldsymbol{z}$ since we only work with the selected codewords in this second stage. Instead, we use the superscript to indicate the number of Transformer blocks traversed so that $\boldsymbol{z}_{i,c}^l$ correspond to the representation $\boldsymbol{z}_{i,c}$ after $l$ blocks.

### 4.2.2 Auto-regressive setup on two dimensions

Our model uses the codewords vectors $Z$ as input to model the distribution of indices K. Formally, we have:

$$P\left((k_{1,1}..., k_{1,C}), ..., (k_{n,1}, ..., k_{n,C})\right) = \prod_{i=1}^{n} \prod_{c=1}^{C} P(k_{i,c} | \boldsymbol{z}_{<i,1}, ..., \boldsymbol{z}_{<i,C}, \boldsymbol{z}_{i,<c}), \tag{9}$$

where $\boldsymbol{z}_{<i,j}$ refers to all the vectors having the first index smaller than $i$ and $j$ as second index.

We introduce the description of the inputs and outputs of our model as it should help to understand how our auto-regressive model accesses all and only allowed positions.

For the input, we offset the first dimension (i.e., the node dimension) by one position to predict the partition indices of the next node, and we concatenate the known partitions of the current node to predict the next partition. Since the number of known partitions varies, so does the size of the input vectors. Therefore, we linearly project these vectors so that all the input vectors have the same dimension $d_{model}$. So, we have:

$$\boldsymbol{z}_{i,0}^0 = W_0^{in}[\boldsymbol{z}_{i-1,1}, ..., \boldsymbol{z}_{i-1,C}], \tag{10}$$

and

$$\boldsymbol{z}_{i,c}^0 = W_c^{in}[\boldsymbol{z}_{i-1,1}, ..., \boldsymbol{z}_{i-1,C}, \boldsymbol{z}_{i,1}, ..., \boldsymbol{z}_{i,c}], \tag{11}$$

where the $W_c^{in} \in \mathbb{R}^{d_{model} \times (d_{latent} + (c/C)d_{latent})}$ are matrices of parameters and $[,]$ is the concatenation operator. To obtain the vector in the first position $\boldsymbol{z}_{1,c}$, we introduce a virtual node $(\boldsymbol{z}_{0,1}, ..., \boldsymbol{z}_{0,C})$ made of vectors of zeros.

After $L$ layers, we project the output $\boldsymbol{z}^L_{i,c}$ and apply softmax to model the index distribution of the next partition.

$$P_\theta(k_{i,c}) = \text{softmax}(f_{out}(\boldsymbol{z}^L_{i,c-1})). \tag{12}$$

We pad the sequence with an end-of-sequence token so that the sequence can take various sizes. Since we sorted the indices in the training set, the indices lower than the last known index are out of the distribution; for instance, $p(k_{i,1} >= k_{i-1,1}) = 0$. Even though the auto-regressive model should learn to generate the sequence following the order in the training set, we ease the task of our model by enforcing this constraint with masking.

### 4.2.3 Transformer

The Transformer is based on attention. The attention function takes queries $\boldsymbol{q}_i$, keys $\boldsymbol{k}_i$, and values $\boldsymbol{v}_i$, all are transformations from the corresponding representation $\boldsymbol{z}_i$. The attention output is a weighted sum of the values, where the weights correspond to a compatibility function, such as the scalar product, between queries and keys. The queries correspond to the aggregating position, while keys and values correspond to aggregated positions. The original attention from (Vaswani et al., 2017) includes a scaling factor so that the attention output is computed as:

$$\boldsymbol{a}_i = \sum_{j \leq i} \text{softmax}\left(\frac{\boldsymbol{q}_i^T \boldsymbol{k}_j}{\sqrt{d_k}}\right)\boldsymbol{v}_j. \tag{13}$$

In practice, all attention functions are computed simultaneously using matrix multiplication. Masking $M$ ensures that the weighted sum includes allowed positions only. We have: $A = \text{softmax}(\frac{QK^T}{\sqrt{d_k}} \odot M)V$. Following the original paper (Vaswani et al., 2017), we use multi-head attention, residual connections, position-wise fully connected feed-forward networks, and positional encoding.

**2D Attention** The main idea behind 2D-Transformer lies in the computation of the attention function. Instead of computing the attention function for all the partitions, we compute it over the node embeddings as a whole. In addition to the benefits offered by transformers over other auto-regressive models, such as the computational complexity per layer, the parallelizable computation, and the short path lengths between long-range dependencies (Vaswani et al., 2017), our 2D-Transformer implements parameter sharing for computational efficiency and better generalization. So, the keys and values are computed once for the whole node, so that we have: $\boldsymbol{k}^l_i = f^l_k(\boldsymbol{z}^l_{i+1,0})$, and $\boldsymbol{v}^l_i = f^l_v(\boldsymbol{z}^l_{i+1,0})$. One can interpret it as a form of parameter sharing since we use the same keys and values and, therefore, the same set of parameters for all the partitions. Nevertheless, the attention weights depend on the partition through the queries, which are different for each partition, as $\boldsymbol{q}^l_{i,c} = f^l_{q,c}(\boldsymbol{z}^l_{i,c})$. In practice, the functions $f_k$, $f_v$, and $f_q$ are small neural networks.

Thanks to the queries, the attention score, $\boldsymbol{q}^T_{i,c}\boldsymbol{k}_j$, also depends on both dimensions. So, the main change of our model compared to the standard Transformer appears in the attention function, which we compute as:

$$\boldsymbol{a}_{i,c} = \sum_{j < i} \text{softmax}\left(\frac{\boldsymbol{q}^T_{i,c}\boldsymbol{k}_j}{\sqrt{d_k}}\right)\boldsymbol{v}_j. \tag{14}$$

Due to the unique key and value for the first dimension, we compute the summation along the second dimension by including only the previous representations ($j < i$). As the original Transformer, we scale the attention score by $\sqrt{d_k}$, where $d_k$ is the dimensionality of the vector $\boldsymbol{k}$. The information about previous partitions in the current node passes through the residual connections as presented hereunder.

**Transformer blocks** Except for the attention function, our model follows the Transformer architecture. We compute attention in every head of every layer and use element-wise feed-forward neural network, layer normalization, as well as residual connections:

$$\boldsymbol{z}_{i,c}^{l+1} = \text{LayerNorm}(\tilde{\boldsymbol{a}}_{i,c}^{l} + f_z^l(\tilde{\boldsymbol{a}}_{i,c}^{l})), \tag{15}$$

where

$$\tilde{\boldsymbol{a}}_{i,c}^{l} = \text{LayerNorm}(\boldsymbol{z}_{i,c}^{l} + [\boldsymbol{a}_{i,c}^{l}]_1^H), \tag{16}$$

and where $[]_1^H$ is the concatenation of the $H$ heads. After $L$ blocks, we take the outputs $\boldsymbol{z}_{i,c}^{l+1}$ to model the probability distribution of the next index, as described in Equation 12.

### 4.3 Generation

For generation, we iteratively sample from $p_\theta(k_{i,c})$, and get the corresponding $\boldsymbol{z}_{i,c}$. Sequence generation stops upon sampling the end-of-sequence token or reaching the maximum number of node embeddings $n_{max}$. So, we can generate graphs of various sizes, and we need at most $n_{max}C$ iterations to generate an instance.

Despite the auto-regressive method, generation with our model is significantly faster than other methods. There are three reasons for this. First, at each iteration, the computational complexity is bounded by the vector-matrix product $q_{i,c}^T K^T$, which is $\mathcal{O}(d_{model}^2)$, where most models require at least $\mathcal{O}(n^w)$, where $w$ is the matrix multiplication exponent. Second, the length of the auto-regressive sequence for graphs usually depends on the number of (potential) edges and is proportional to $n^2$. Instead, our method is auto-regressive over a sequence depending linearly on $n$. For the current size of the generated graphs, $nC$ is also smaller than the number of time steps in score-based models. We input the generated sets into the decoder and generate the resulting graphs simultaneously. Finally, the efficiency of our method allows us to keep the size of our model small. As presented in Appendix A.3, the largest hidden dimension of our experimental models never exceeds 256 hidden units, far from the dimension of current large models. In Section 5, we experimentally show that our method is comparatively fast at generation.

## 5 Evaluation

This section presents the results of our experiments evaluating our model. We first compare the performance of our DGAE with baseline models, both on simple and annotated graphs. We then perform various ablation studies to evaluate and understand our model better.

### 5.1 Experimental setup

We evaluate our model on both synthetic datasets and molecular datasets. In the following, we present these datasets, the metrics used to assess the quality of generated samples, and the baseline models used for comparison. Appendix A.3 reports the details of the models, the hyperparameters choices and the resulting number of parameters used for the experiments. For a fair comparison, we adopt the experimental procedure followed in (Jo et al., 2022) regarding metrics, split between training and test sets, number models and runs, and sample sizes described hereafter.

#### 5.1.1 Simple graphs

For simple graphs, we evaluated the performance of our model on two datasets of small graphs, namely Ego-Small and Community-Small, with 200 and 100 graphs, respectively as well as Enzymes, a dataset of larger graphs. In particular, we split the datasets between training and test sets with a ratio of $80\% - 20\%$ as (Jo et al., 2022) and use the exact same split when available.

The Ego-Small dataset is constructed by extracting ego-networks from the large citeseer network, a real-world social network. The dataset comprises 200 graphs having between 4 and 18 nodes.

The Community-Small dataset is purely synthetic. The number of nodes in each graph is uniformly sampled from the set $\{12, 14, 16, 18, 20\}$. Each graph is divided into two communities of equal size. The probability of edge intra-community is set to $p_{intra} = 0.7$, while the probability of edge between inter-communities is set

to $p_{inter} = 0.03$, with the constraint of having at least one edge between communities. We use the version of the dataset proposed by (Jo et al., 2022) containing 100 graphs.

Enzymes is a dataset containing 587 protein graphs that represent the protein tertiary structures of enzymes. It is extracted from the BRENDA database. The graphs in this dataset are larger than in Ego-small and Community-Small, with a node range between 10 and 125.

We employ the maximum mean discrepancy (MMD) to compare the distributions of graph statistics between generated and test graphs (You et al., 2018). The MMDs are computed over the distributions of degrees (deg.), clustering coefficients (clust.), and the number of occurrences of orbits with up to four nodes (orbit)[3]. Similar to (Jo et al., 2022), we utilize the Gaussian Earth Mover's Distance (EMD) kernel to compute the MMDs.

The MMDs are computed by comparing the test set to generated samples of the same size as the test set. For Ego-Small and Community-Small, our reported results are the average of fifteen runs, i.e., fifteen generation batches with the test set size used for evaluation against the test set: three runs from five models trained independently. For the Enzymes dataset, which contains much larger graphs, we follow (Jo et al., 2022) and average over three runs from a single model. Due to space limitation, we report the standard deviation in the appendix B.

### 5.1.2 Molecular datasets

We evaluated the performance of our model on two widely used datasets for molecule generation: Qm9 (Ramakrishnan et al., 2014) and Zinc250k (Irwin et al., 2012).

The Qm9 dataset (Ramakrishnan et al., 2014) comprises 133,885 small organic molecules with up to nine heavy atoms, consisting of only four atom types, namely $C, O, N, F$. We followed the convention in previous works and employed the kekulized version of the dataset with three edge types (single, double, and triple).

The Zinc250k dataset is a subset of the Zinc database (Irwin et al., 2012) that includes 250,000 molecules with up to 38 heavy atoms of nine types. We also used the kekulized representation of this dataset.

We used the test sets provided by (Jo et al., 2022). The metrics are calculated over 10,000 samples from training and test sets when needed.

Traditionally, molecule generation was evaluated through three metrics: validity, uniqueness, and novelty. We argue that these metrics are unsuitable for evaluating the generative graph models.

Recent works mostly use valency corrections, bringing the validity rate to 100%. Consequently, this metric has become uninformative. Uniqueness and novelty primarily indicate potential mode collapse and overfitting, respectively. Since most models, including ours, achieve a high score, these metrics provide little information for evaluation. Therefore, we do not use these metrics for evaluation. Nonetheless, we report them in the Appendix B for completeness.

Instead, we use the Fréchet ChemNet Distance (FCD) (Preuer et al., 2018) and the Neighborhood subgraph pairwise distance kernel (NSPDK) MMD (Costa & Grave, 2010) metrics. As highlighted in (Jo et al., 2022) *'FCD and NSPDK MMD are salient metrics that assess the ability to learn the distribution of the training molecules, measuring how close the generated molecules lie to the distribution'*. FCD assesses the generated molecules in chemical space, while NSPDK MMD evaluates the distribution of the graph structures.

In addition to FCD and NSPDK metrics, we also include the validity rate without correction as a supplementary evaluation metric. This metric calculates the fraction of valid molecules without any valency correction or edge resampling, and we employ the version that allows for formal charges as in (Jo et al., 2022).

---

[3]We remind that an orbit is the equivalence class of nodes of a graph under the action of its automorphisms. Less formally, it is a set of nodes that are structurally similar. In Figure 13 of appendix D, the nodes with the same color belong to the same orbit

## 5.2 Baseline models

We present the results of six baseline models in our experiments for simple graphs: three sequential and three permutation-equivariant models. The first model, GraphRNN (You et al., 2018), is an RNN-based auto-regressive model and is the paper that introduced the experiments and metrics we use. The second model, GraphDF (Luo et al., 2021), is a discrete flow-based model, and GraphARM (Kong et al., 2023) is the recent diffusion-based sequential model. The third and fourth models are one-shot continuous diffusion models, namely EDP-GNN (Yang et al., 2019) and GDSS (Jo et al., 2022). EDP-GNN is initially designed for simple graphs, while GDSS can generate both simple and annotated graphs. Finally, DiGress (Vignac et al., 2023) is a discrete diffusion model.

In addition to these models, we randomly sample the same number of graphs from the training set as in the test set to use as a baseline. It gives us the distance between two random samples from the true distribution. Models under this baseline indicate that the generated sample overfits the test set. We report the average MMDs between the test set and 15 random samples drawn from the training set.

For molecular graphs, we also present the results of six baseline models. We utilize GraphAF, GraphDF (Luo et al., 2021) and GraphARM. Additionally, we compare our model against MoFlow (Zang & Wang, 2020), a one-shot flow-based model, EDP-GNN (Yang et al., 2019) a score-based model adapted by (Jo et al., 2022) to handle annotated graphs, GDSS(Jo et al., 2022), another score-based model and DiGress (Vignac et al., 2023), a discrete diffusion model. We take the results from (Jo et al., 2022) for all baselines except GraphARM and DiGress taken from (Kong et al., 2023).

As discussed in Section 3, there are models specifically designed for graph generation using different types of representation. In particular, some recent models, such as E(3) Equivariant Diffusion Model (EDM) and Geometric Latent Diffusion Models (GEOLDM) leverage diffusion models representing molecules as 3D coordinates point clouds. These models predict the connections between atoms *post hoc* based on the distance between atoms. Compared to our experiments, these models use additional domain-knowledge information, particularly the atom positions. So, they must use datasets including this information, such as Qm9, but excluding Zinc250k. Moreover, they use specific metrics such as atom and molecule stability. Conversely, they do not report specific metrics for graphs such as NSPDK. Consequently, the only point of comparison is the validity rate on QM9. On this metric, EMD and GEOLDM obtain close and comparable results with the models used as baselines for our experiment with 91.9% and 93.8%, respectively.

## 5.3 Results

This section reports a quantitative evaluation of the experiments. We provide visualization of the generated graphs in Appendix C.

We give the evaluation results on Ego-Small and Community-Small datasets in Table 1. In the case of Ego-Small, which is a relatively small and straightforward dataset, we observe that our model matches the scores obtained with random samples from the training set. GDSS and DiGress (Jo et al., 2022; Vignac et al., 2023) also reaches this level (or better!) on all metrics. Therefore, this dataset does not allow to discriminate between the best models.

On the Community-Small dataset, our model presents the best average results compared to baseline models. Table 2 presents the results on the Enzymes datasets. Again, our model outperforms baseline models on average.

### 5.3.1 Molecular graphs

As presented in Table 3, DiGress outperforms other models over all metrics in the QM9 dataset. However, it performs much worst on the larger molecules of the Zinc250k dataset. Except DiGress on QM9, our DGAE model exhibits superior performance to all other models regarding FCD and NSPDK metrics, frequently surpassing them by a substantial margin. However, we observe that it is slightly under the best models regarding the validity without correction metric while remaining competitive. These results suggest that, compared to baseline models, our model performs better at capturing global graph features as reported

Table 1: MMDs distances based on degrees (deg.), clustering coefficients (clust.), and the number of occurrences of orbits with four nodes (orbit) and their averages (avg.) on datasets of small, simple graphs.

| | | Ego-Small | | | | Community-Small | | | |
|---|---|---|---|---|---|---|---|---|---|
| | **Model** | **Deg.↓** | **Clust.↓** | **Orbit↓** | **Avg↓** | **Deg.↓** | **Clust.↓** | **Orbit↓** | **Avg↓** |
| | Training set | 0.022 | 0.043 | 0.0062 | 0.024 | 0.015 | 0.026 | 0.0032 | 0.015 |
| Auto-reg. | GraphRNN | 0.090 | 0.220 | **0.003** | 0.104 | 0.080 | 0.120 | 0.040 | 0.080 |
| | GraphDF | 0.04 | 0.13 | 0.01 | 0.060 | 0.06 | 0.12 | 0.03 | 0.070 |
| | GraphARM | 0.019 | 0.017 | 0.010 | **0.015** | 0.034 | 0.082 | **0.004** | 0.040 |
| One-shot | EDP-GNN | 0.052 | 0.093 | 0.007 | 0.051 | 0.053 | 0.144 | 0.026 | 0.074 |
| | GDSS | 0.021 | **0.024** | 0.007 | 0.017 | 0.045 | 0.086 | 0.007 | 0.046 |
| | DiGress | **0.015** | 0.029 | 0.005 | 0.016 | 0.047 | **0.041** | 0.026 | 0.038 |
| Ours | DGAE | 0.021 | 0.041 | 0.007 | 0.023 | **0.032** | 0.062 | 0.0046 | **0.033** |

Table 2: MMDs distances based on degrees (deg.), clustering coefficients (clust.), and the number of occurrences of orbits with up to four nodes (orbit) and their averages (avg.) on the Enzymes dataset.

| | | Enzymes | | | |
|---|---|---|---|---|---|
| | **Model** | **Deg.↓** | **Clust.↓** | **Orbit↓** | **Avg↓** |
| | Training set | 0.004 | 0.021 | 0.001 | 0.008 |
| Auto-reg. | GraphRNN | 0.017 | 0.062 | 0.046 | 0.042 |
| | GraphDF | 1.503 | 1.061 | 0.202 | 0.922 |
| | GraphARM | 0.029 | 0.054 | 0.015 | 0.033 |
| One-shot | EDP-GNN | 0.052 | 0.093 | 0.007 | 0.051 |
| | GDSS | 0.026 | 0.061 | 0.009 | 0.032 |
| | DiGress | **0.004** | 0.083 | **0.002** | 0.030 |
| Ours | DGAE | 0.020 | **0.051** | 0.003 | **0.025** |

by the FCD and the NSPDK metrics on larger molecules. It remains competitive at capturing the local node-edge interactions necessary to generate valid molecules, even though some baseline models are better in this respect.

Table 3: MMD distances based Neighborhood subgraph pairwise distance kernel (NSPDK) and Fréchet ChemNet Distance on the Qm9 and Zinc250k datasets.

| | Model | Qm9 | | | Zinc | | |
|---|---|---|---|---|---|---|---|
| | | NSPDK↓ | FCD↓ | Valid. %↑ | NSPDK↓ | FCD↓ | Valid. %↑ |
| auto-reg. | GraphAF | 0.021 | 5.53 | 74.4 | 0.044 | 16.0 | 68.5 |
| | GraphDF | 0.064 | 10.92 | 93.88 | 0.177 | 33.5 | 90.6 |
| One-shot | MoFlow | 0.017 | 4.47 | 91.4 | 0.046 | 20.9 | 63.1 |
| | EDP-GNN | 0.005 | 2.68 | 47.5 | 0.049 | 16.7 | 83.0 |
| | GDSS | 0.003 | 2.90 | 95.7 | 0.019 | 14.7 | **97.0** |
| | DiGress | **0.0005** | **0.36** | **99.0** | 0.082 | 23.1 | 91.0 |
| Ours | DGAE | 0.0015 | 0.86 | 92.0 | **0.007** | **4.4** | 77.9 |

### 5.3.2 Generation time

Finally, we use the Zinc dataset to assess our model's generation time. We compare it against the best auto-regressive model (Luo et al., 2021) and the best one-shot models GDSS and DiGress (Jo et al., 2022; Vignac et al., 2023). We compute the clock time to generate 1000 graphs in the rdkit format on one GeForce RTX 3070 GPU and 12 CPU cores. We report the result in Table 4.

Our model not only presents very high performances at modeling graph distribution, but it is also several orders of magnitude faster at generation than the baseline models .

Table 4: Generation time in seconds to generate 1000 graphs with models trained on molecular datasets .

| | Time (s) | |
|---|---|---|
| Models | Qm9 ↓ | Zinc ↓ |
| GraphDF | 3791.67 | 3859.23 |
| GDSS | 28.07 | 300.44 |
| DiGress | 54.01 | 799.43 |
| DGAE (Ours) | **0.33** | **1.80** |

### 5.4 Ablation study

We run two types of ablation studies. First, we empirically observe the effect of the various types of data augmentation. We then evaluate the effect of the codebooks and partition size. For all the ablation studies, we use the Zinc250k dataset, which is by far the largest dataset used for the experiments (regarding the number of graphs).

### 5.4.1 Graph Features Augmentation

Numerous feature augmentation strategies for graphs have been proposed to overcome the inherent constraints of GNNs (see Section 4.1.2 and 2.4). Using such feature augmentations for graph generation has become a prevalent practice. However, we are not aware of any work evaluating these methods in generative modeling. The primary objective of these auxiliary features is to enhance the identification of graph features. A convenient way of evaluating a model's capability of capturing graph features and substructures is by assessing its reconstruction accuracy. Thus, we evaluate the methods for synthetic graph feature methods based on their reconstruction performance.

Therefore, we have conducted a comparative analysis of several feature augmentation techniques, including spectral embedding, cycle counts, random features, and our $p$-paths method. The impact of each augmentation is evaluated independently against a scenario where no augmentation is applied. Further, we assess the impact of omitting one method compared to the combined use of all techniques. In figure 3, we report the reconstruction loss over 20 epochs.

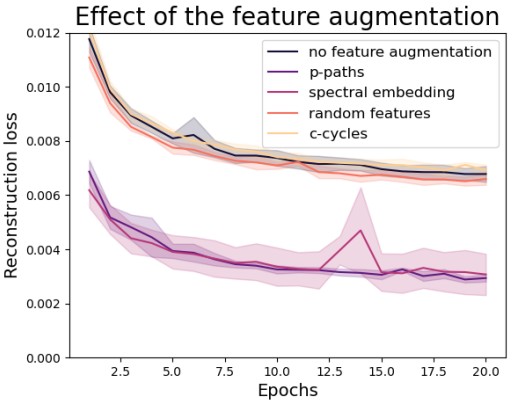 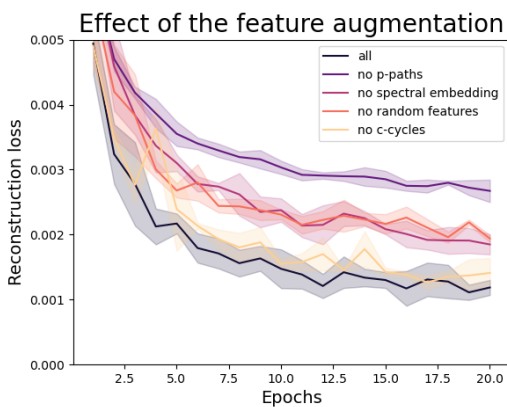

Figure 3: Left: The effect of each feature augmentation scheme on the reconstruction loss compared to the absence of feature augmentation. Right: The effect of removing each feature augmentation scheme compared to using all of them. The lines represent the average over three runs, and the shaded area is the standard deviation.

When used alone, the reconstruction loss curves suggest that the spectral embedding and our $p$-paths method dramatically improve the reconstruction loss. Conversely, the cycle count and random feature techniques demonstrate negligible, if any, effects. Upon removal of a single method, the performance degradation is most noticeable in the absence of our $p$-path method. The effect is also evident for the random and spectral features. Conversely, the cycle count contributes marginally, if at all, to the overall effect. We conclude that our feature augmentation method is effective at enhancing MPNN as a feature extractor on the node neighborhood, showing competitive performance against the computationally expensive method using spectral features.

In table 5, we present the impact of the feature augmentation on the pre-processing time and the training times. We measure the pre-processing time, i.e., the time to transform the dataset stored as a numpy object to a pytorch-geometric dataset, including the augmented features. We also compute the time for 1000 training iterations, i.e., 1000 parameter updates. For both, we compare our $p - path$ method alone against the dataset without feature augmentation and with the spectral feature. We report the clock time measured on the Zinc250k dataset with one GeForce RTX 3070 GPU and 12 CPU cores.

Table 5: Pre-processing and training time for some feature augmentation methods in seconds. We compute the training time over 1000 training iterations.

| Methods | Time (s) | |
|---|---|---|
| | Pre-processing ↓ | Training ↓ |
| No augmentation | 74.81 | 38.19 |
| Spectral feature | 151.22 | 38.09 |
| $P-$path feature (ours) | 231.18 | 38.71 |

We observe that the pre-processing time for our $p-$path scheme is significantly slower than the pre-processing without feature augmentation and the method based on spectral features. Nevertheless, the method remains short in comparison to the training time of graph generative models, which typically spans hours, if not days. Regarding the training, all methods are very close, our $p-$path being marginally slower.

### 5.4.2   Influence of Codebook Sizes and Partitioning

This section evaluates the impact of varying the codebook and partitioning sizes. We first evaluate their effect on dictionary usage. We then study the effects on the autoencoder training *and* the prior training, given that the size of the codebook and the partitioning determine both the length of the auto-regressive sequence $(nC)$ and the dimensionality of the categorical distribution[4] $(m+1)$. For these experiments run on the Zinc250k dataset, we compare ten configurations for various codebook size $m$ and number of partition $C$, representing three sizes of possible codeword sequences $M$: $256^1 = 16^2 = 4^4$, $1024^1 = 32^2 \approx 10^3$, and $4096^1 = 64^2 = 16^3 = 8^4$.

We first analyze the dictionary usage, reporting the normalized perplexity. The perplexity is a measure of the dictionary usage computed as $e^{H(P(z_i^q))}$, where $H(P(z_i^q))$ is the entropy over the dictionary likelihood. The closer the distribution is to the uniform distribution, the higher the perplexity. The maximum of perplexity corresponds to the dictionary size $M$. To compare various dictionary sizes, we use the normalized perplexity $\frac{1}{M}e^{H(P(z_i^q))}$. We present the result in Figure 4. We see that the normalized perplexity slightly decreases as the dictionary size increases. We also see that using a single large codebook, in this case, a single codebook of size 4096, leads to codebook collapse.

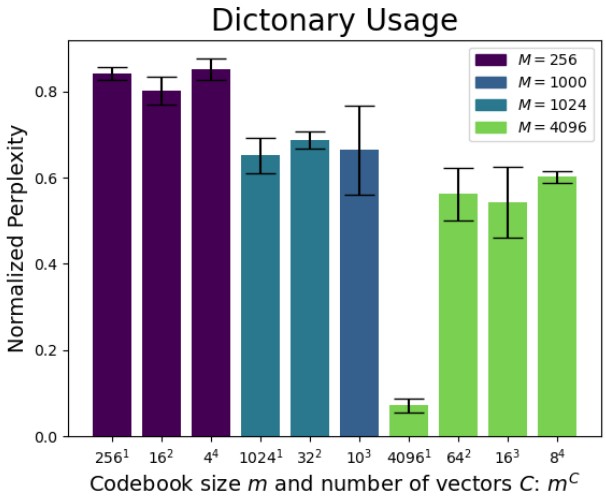

Figure 4: effect of the codebook size and the partitioning on the dictionary usage. We report the normalized perplexity averaged over three runs. The black lines indicate the standard deviations.

We then assess the impact of the codebook size $m$, the number of partitions $C$, and the dictionary size (i.e., the number of possible codeword sequences $M = m^C$) on the reconstruction loss of the auto-encoder. We analyze the same ten configurations. For comparison, we also include an experiment without quantization. We then selected the best reconstruction loss on the test set (calculated after each epoch) and reported the average across three runs. Figure 5 depicts the outcome. Our findings suggest that the number of possible sequences is critical, particularly when small (256 in our experiment). Conversely, the absence of quantization yields almost perfect reconstruction (so that its representation in Figure 5 is barely perceptible). However, the benefits of increasing the number of possible sequences become limited once its number surpasses a certain threshold. In this experiment, the advantage of increasing the number of possible sequences from 1024 to 4096 is unclear. Likewise, the partitioning effects become significant when the number of possible sequences is low, with smaller partitions proving more beneficial. However, the impact decreases with larger numbers of possible sequences. In our experiments, we do not see any effect when increasing the number of possible sequences to 4096. In Appendix C, we present similar figures reporting node and edge error rates. With a dictionary size larger than 256, we reach close to perfect reconstruction, with both error rates lying below 0.001 for almost all configurations.

---

[4]The '+1' refers to the end-of-sequence token

We finally observe the effect on generation. We do not present the result without quantization because our 2D-Transformer requires quantized vectors for training. Therefore, our model is not able to perform generation without quantization. We report here the NSPDK metric, but the FCD, reported in Appendix C, presents similar results. The benefits of a large number of arrangements and a small number of partitions vanish. The most significant observation is the poor result for $m = 4096$ and $C = 1$. We assume that the drop in performance comes from codebook collapse observed in Figure 4, which results in a high probability of sampling indices out of the distribution. All the other settings result in comparable NSPDK values, which are lower than those of the baseline models. We also observe that the configuration with two partitions ($C = 2$) and a codebook size of $m = 32$ presents slightly better generation metrics than the other configurations.

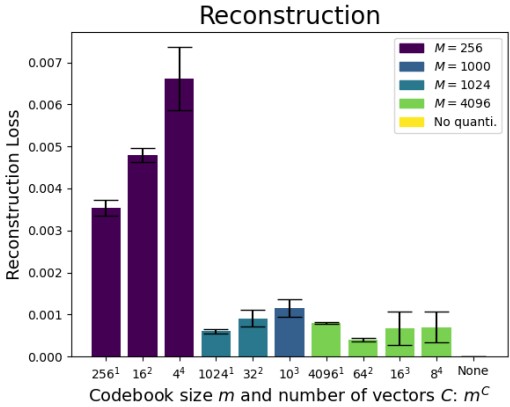 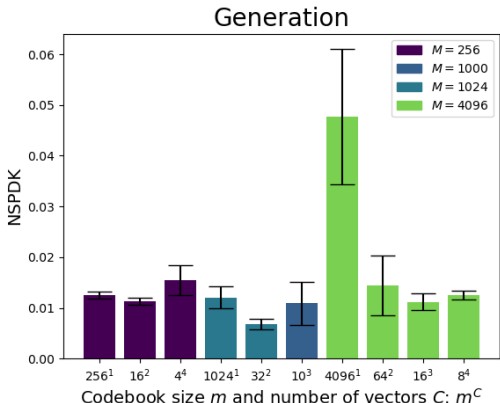

Figure 5: effect of the codebook size and the partitioning on reconstruction (left) and generation (right). We report the best reconstruction loss and the best NSPDK averaged over three runs. The black lines indicate the standard deviations.

## 6 Conclusion

Our work introduces DGAE, a powerful generative model for graphs in two stages. In the first stage, we leverage an auto-encoder, mapping graphs to sets and recovering graphs from sets. We use permutation-equivariant functions to prevent the difficulties caused by the multiple possible representations of graphs. In the second stage, we sort the sets into sequences and learn them auto-regressively. In its present stage, we do not see any ethical concerns with our model.

Our model achieves excellent performance at learning graph distributions. Our empirical evaluation shows that it is competitive with the best baseline models. Moreover, our model is an order of magnitude faster at generation than competitive baseline models.

Implementing our model, we introduce two innovative techniques: a new feature augmentation method and a new Transformer architecture for sequential data over two dimensions.

However, our model also presents some limitations. First, our model requires two stages of training. This makes the hyperparameter tuning complicated, particularly for hyperparameters that influence both training stages, such as the number of partitions $C$ or the codebook size $m$, for instance. Second, our model cannot perform conditional generation yet. We hypothesize that conditioning the auto-regressive model might be sufficient. Nevertheless, the elaboration and validation of conditional generation remain topics for future investigation.

Lastly, addressing the challenge of scaling models to accommodate larger graphs is a prevalent concern in generative modeling, one that our model does not currently tackle. We have evaluated our model on graphs with up to 125 nodes. Adapting our model to handle larger graphs would raise challenges related to computation, memory, and overall model performance. This is another line of research open for further exploration.

Distinct from the two predominant approaches in current literature, i.e., the fully sequential and the fully permutation-equivariant methods, our generative model for graphs introduces an innovative and competitive framework. We view its present limitations as promising opportunities for future research.

### Acknowledgments

We acknowledge the financial support of the Swiss National Science Foundation within the LegoMol project (grant no. 207428). The computations were performed at the University of Geneva on "Baobab" and "Yggdrasil" HPC clusters.

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
