# A Model Implementation Details

In this section, we provide more details on the implementation of our model used for the experiments.

## A.1 Decoder Output

For simple graphs, the edge representation $e_{i,j}^L$ after the last layer $L$ is a scalar, which we pass through a sigmoid function $\sigma$. The result is interpreted as the probability of the presence of an edge $\sigma(e_{i,j}^L) = p_\theta(\epsilon_{i,j} = 1|\mathcal{Z}^q)$. In the case of annotated graphs with discrete attributes, the outputs corresponding to each node $\boldsymbol{x}_i^L \in \mathbb{R}$ and each edge each edge $\boldsymbol{e}_{i,j}^L \in \mathbb{S}$ are passed through a softmax function $\sigma$, yielding the probability distribution over their attributes, i.e., softmax($\boldsymbol{e}_{i,j}^L) = p_\theta(\boldsymbol{e}_{i,j}|\mathcal{Z}^q)$ and softmax($\boldsymbol{x}_i^L) = p_\theta(\boldsymbol{x}_i|\mathcal{Z}^q)$. Note that for edges, the absence of an edge has to be encoded as one of the output category.

Our model output different value for $\boldsymbol{e}_{i,j}$ and $\boldsymbol{e}_{j,i}$. So, we can model directed graph. For undirected graphs, we enforce the symmetry of the adjacency by averaging the output matrix by its transpose.

For sampling, we always chose the mode of the discrete distribution $\hat{x}_i = \text{argmax}_{\{r \in \mathbb{N}: r \leq R\}}(p_\theta(x_{i,r}|\mathcal{Z}^q))$ and $\hat{e}_{i,j} = \text{argmax}_{\{s \in \mathbb{N}: s \leq S\}}(p_\theta(e_{i,j,s}|\mathcal{Z}^q))$.

## A.2 Auto-encoder Training

**Codebooks initialization** The codebook initialization is important for the training quality of the auto-encoder. As proposed by Łańcucki et al. (2020), we initiate the auto-encoder training for $T_{init}$ steps without quantization. Next, we collect $S$ node embeddings, $S$ being an hyperparameter, and perform $k$-means++ clustering. We use the resulting vectors as initial codebook vectors. We use $S = 100K$ samples. Unlike Łańcucki et al. (2020), we do not re-estimate the codebook vector periodically. After initialization, we continue to update the codebooks thanks to a dedicated loss function details in Section 4.1.5.

**Reconstruction loss** We define the reconstruction loss as the negative log-likelihood. In practice, we use the binary cross-entropy loss for simple graphs and the cross-entropy loss for annotated graphs.

Also, for annotated graphs, the respective weight of the nodes and the edges in the loss function is an hyperparameter choice. In practice, for all our experiments, we use a constant weight for the edges and the nodes. For instance, for a graph with categorical distributions over nodes and edges, we used:

$$\mathcal{L}_{\text{recon.}} = \frac{1}{n+n^2}\left(\sum_{i=1}^{n}\sum_{r=1}^{R}x_{i,r}\ln(\tilde{x}_{i,r}) + \sum_{i=1}^{n}\sum_{\substack{j=1\\j\neq i}}^{n}\sum_{s=1}^{S}e_{i,j,s}\ln(\tilde{e}_{i,j,s})\right), \tag{17}$$

where $R$ and $S$ are distribution supports of the node and the edge, respectively. The reconstruction loss can be easily modify to fit multiple variables per node and edge.

**Commitment loss** To prevent the expansion of the encoder outputs, we incorporate a regularization, similar to van den Oord et al. (2017), which keeps the learnt representations close to the cluster centers. We define it as the mean square distance between the partition vector and its corresponding codeword:

$$\mathcal{L}_{commit.} = \frac{1}{nC}\sum_{i=1}^{n}\sum_{c=1}^{C}||\boldsymbol{z}_{i,c}^h - \text{sg}[\boldsymbol{z}_{i,c}^q]||_2^2. \tag{18}$$

### A.3 Hyperparameters

Table 6: Parameters shared across experiments.

| Data | Features augmentation | all |
|---|---|---|
| | layers in the neural networks | 3 |
| GNNs | activation function | ReLu |
| | Size of the hidden representations $\boldsymbol{e}_{i,j}^l$ and $\boldsymbol{x}_i^l$ | 32 |
| | partitions $C$ | 2 |
| Quantizer | latent vector size | 8 |
| | $\beta$ (loss commitment cost) | 0.25 |
| | $\gamma$ (loss weight) | 0.1 |
| Training | betas (adam optimizer) | 0.9, 0.99 |
| | learning rate decay | 0.5 |
| | heads in multihead att. | 16 |
| 2d-transfomer | layers in the neural networks | 4 |
| | units in the hidden layers of the mlp's | $2 \times d_{models}$ |
| | activation function | ReLu |

Table 7: Parameters for the various experiments.

| | dataset | Zinc | Qm9 | Ego | Com-ity | Enzymes |
|---|---|---|---|---|---|---|
| GNNs | number of GNN layers | 4 | 4 | 2 | 2 | 6 |
| | units in mlp's hidden layers | 128 | 128 | 64 | 64 | 128 |
| Quantizer | codebook size $K$ | 32 | 16 | 8 | 16 | 32 |
| | initialization steps | 1000 | 1000 | 0 | 0 | 100 |
| Training | batch size | 32 | 32 | 32 | 32 | 16 |
| | param. updates between decay | 25k | 25k | 10k | 10k | 10k |
| 2d-transformer | blocks | 6 | 6 | 3 | 3 | 6 |
| | $d_{model}$ | 256 | 128 | 64 | 64 | 128 |

Table 8: Number of parameters in our experimental models

| Datasets | Zinc | Qm9 | Ego | Community | Enzymes |
|---|---|---|---|---|---|
| Auto-encoder | 749K | 746K | 107k | 107 | 1.14M |
| 2D-Transformer | 75.4M | 18.9M | 2.47M | 2.47M | 18.9M |
| Total | 76.2M | 19.7M | 2.58M | 2.58M | 20.1M |

## B   Detailed results

All results in this section, except DGAE, are taken from Jo et al. (2022). Unfortunately, the results for GraphARM and DiGress are taken from Kong et al. (2023), which do not produce the standard deviation. So, we do not have more information than the ones produced in the core of the text.

### B.1   Qm9

Results are the means and the standard deviations of 3 runs.

Table 9: Generation results on the **Qm9** dataset.

| Model | NSPDK↓ | FCD↓ | Val. wo. corr. %↑ |
|---|---|---|---|
| GraphAF | 0.021 ± 0.003 | 5.53 ± 0.40 | 74.4 ± 2.6 |
| GraphDF | 0.064 ± 0.000 | 10.92 ± 0.0 | 93.8 ± 4.8 |
| GraphARM | 0.002 | 1.22 | 90.3 |
| MoFlow | 0.017 ± 0.003 | 4.47 ± 0.60 | 91.4 ± 1.2 |
| EDP-GNN | 0.005 ± 0.001 | 2.68 ± 0.22 | 47.5 ± 3.6 |
| GDSS | 0.003 ± 0.000 | 2.90 ± 0.28 | 95.7 ± 0.8 |
| DiGress | **0.0005** | **0.36** | **99.0** |
| DGAE | 0.0015 ± 0.0000 | 0.86 ± 0.02 | 92.0 ± 0.25 |

Table 10: Generation results on the **Qm9** dataset.

| Model | Uniqueness↓ | Novelety.↓ | Validity %↑ |
|---|---|---|---|
| GraphAF | 88.64 ± 2.37 | 86.59 ± 1.95 | 100.00 ± 0.00 |
| GraphDF | 98.58 ± 0.25 | 98.54 ± 0.48 | 100.00 ± 0.00 |
| GraphARM | | | |
| MoFlow | 98.65 ± 0.25 | 94.72 ± 0.77 | 100.00 ± 0.00 |
| EDP-GNN | 99.25 ± 0.05 | 86.58 ± 1.85 | 100.00 ± 0.00 |
| GDSS | 98.46 ± 0.61 | 86.27 ± 2.29 | 100.00 ± 0.00 |
| DiGress | | | |
| DGAE | 97.61 ± 0.17 | 79.09 ± 0.42 | 100.00 ± 0.00 |

## B.2  Zinc

Results are the means and the standard deviations of 3 runs.

Table 11: Generation results on the **Zinc** dataset.

| Model | NSPDK↓ | FCD↓ | Val. wo. corr. %↑ |
|---|---|---|---|
| GraphAF | 0.044 ± 0.005 | 16.0 ± 0.5 | 68.5 ± 1.0 |
| GraphDF | 0.177 ± 0.001 | 33.5 ± 0.2 | 90.6 ± 4.3 |
| MoFlow | 0.046 ± 0.002 | 20.9 ± 0.2 | 63.1 ± 5.2 |
| EDP-GNN | 0.049 ± 0.006 | 16.7 ± 1.3 | 83.0 ± 2.7 |
| GDSS | 0.019 ± 0.001 | 14.7 ± 0.7 | 97.0 ± 0.8 |
| DGAE | 0.007 ± 0.000 | 4.4 ± 0.0 | 77.9 ± 0.5 |

Table 12: Generation results on the **Zinc** dataset.

| Model | Uniqueness↓ | Novelty↓ | Validity %↑ |
|---|---|---|---|
| GraphAF | 98.64 ± 0.69 | 99.99 ± 0.01 | 100.00 ± 0.00 |
| GraphDF | 99.63 ± 0.01 | 100.00 ± 0.00 | 100.00 ± 0.00 |
| MoFlow | 99.99 ± 0.01 | 100.00 ± 0.00 | 100.00 ± 0.00 |
| EDP-GNN | 99.79 ± 0.08 | 100.00 ± 0.00 | 100.00 ± 0.00 |
| GDSS | 99.64 ± 0.13 | 100.00 ± 0.00 | 100.00 ± 0.00 |
| DGAE | 99.94 ± 0.03 | 99.97 ± 0.01 | ±100.00 ± 0.00 |

### B.3 Ego-small

Results are the means and the standard deviations of 15 runs, 3 different runs for 5 independently trained models.

Table 13: Generation results on the **Ego-Small** datasets.

| Model | Degrees↓ | Cluster.↓ | Orbits↓ |
|---|---|---|---|
| GraphRNN | 0.090 | 0.220 | **0.003** |
| GraphDF | 0.04 | 0.13 | 0.01 |
| EDP-GNN | 0.052 | 0.093 | 0.007 |
| GDSS | $0.021 \pm 0.008$ | $0.024 \pm 0.007$ | $0.007 \pm 0.005$ |
| DGAE (Ours) | $0.021 \pm 0.010$ | $0.041 \pm 0.026$ | $0.007 \pm 0.005$ |

### B.4 Community-small

Results are the means and the standard deviations of 15 runs, 3 different runs for 5 independently trained models.

| Model | Degrees↓ | Cluster.↓ | Orbits↓ |
|---|---|---|---|
| GraphRNN | 0.080 | 0.120 | 0.040 |
| GraphDF | 0.06 | 0.12 | 0.03 |
| EDP-GNN | 0.053 | 0.144 | 0.026 |
| GDSS | $0.045 \pm 0.028$ | $0.086 \pm 0.022$ | $0.007 \pm 0.004$ |
| DGAE (Ours) | $0.032 \pm 0.019$ | $0.062 \pm 0.032$ | $0.0046 \pm 0.004$ |

Table 14: Generation results on the **Community-Small** dataset.

### B.5 Enzymes

Results are the means and the standard deviations of 3 runs.

Table 15: Generation results on **Enzymes** datasets.

| Model | Deg.↓ | Clust.↓ | Orbit↓ |
|---|---|---|---|
| GraphRNN | **0.017** $\pm 0.007$ | $0.062 \pm 0.020$ | $0.046 \pm 0.031$ |
| GraphDF | $1.503 \pm 0.011$ | $1.061 \pm 0.011$ | $0.202 \pm 0.002$ |
| EDP-GNN | $0.023 \pm 0.012$ | $0.268 \pm 0.164$ | $0.082 \pm 0.078$ |
| GDSS | $0.026 \pm 0.008$ | $0.061 \pm 0.010$ | $0.009 \pm 0.005$ |
| DGAE (Ours) | $0.020 \pm 0.004$ | **0.051**$\pm 0.017$ | **0.003** $\pm 0.001$ |

### B.6 Generation time

Results are the means and the standard deviations of 3 runs.

Table 16: Generation time on molecular datasets in seconds.

| | Time (s) | |
| --- | --- | --- |
| Model | Qm9 ↓ | Zinc ↓ |
| GraphDF | 3791.67 ± 16.21 | 3859.23 ± 8.34 |
| GDSS | 28.07 ± 0.15 | 300.44 ± 1.94 |
| DiGress | 54.01 ± 1.02 | 799.43 ± 38.22 |
| DGAE (Ours) | **0.33 ± 0.01** | **1.80 ± 0.01** |

## C   Visualization

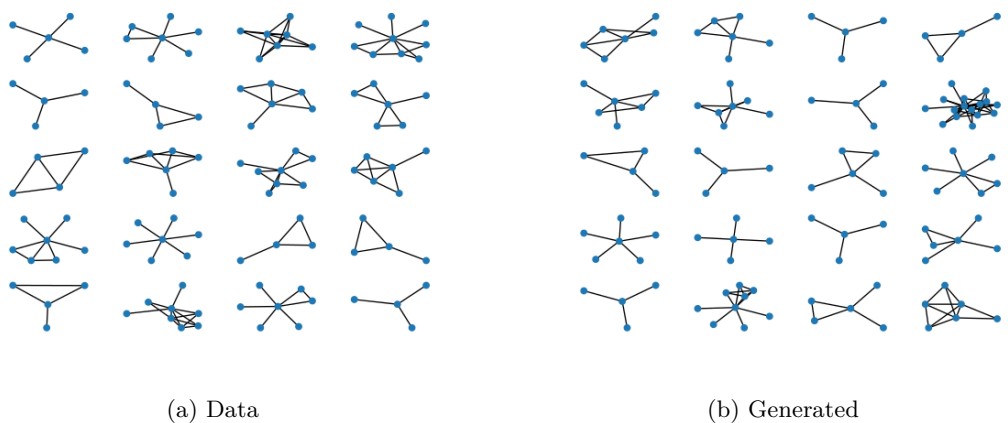

(a) Data                    (b) Generated

Figure 6: Example of graphs from the Ego-Small dataset and from generated samples.

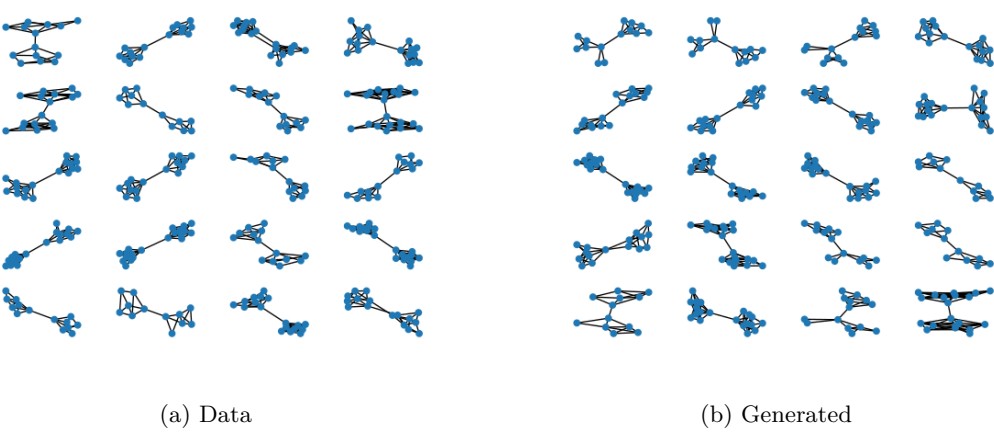

(a) Data                    (b) Generated

Figure 7: Example of graphs from the Community-Small dataset and from generated samples.

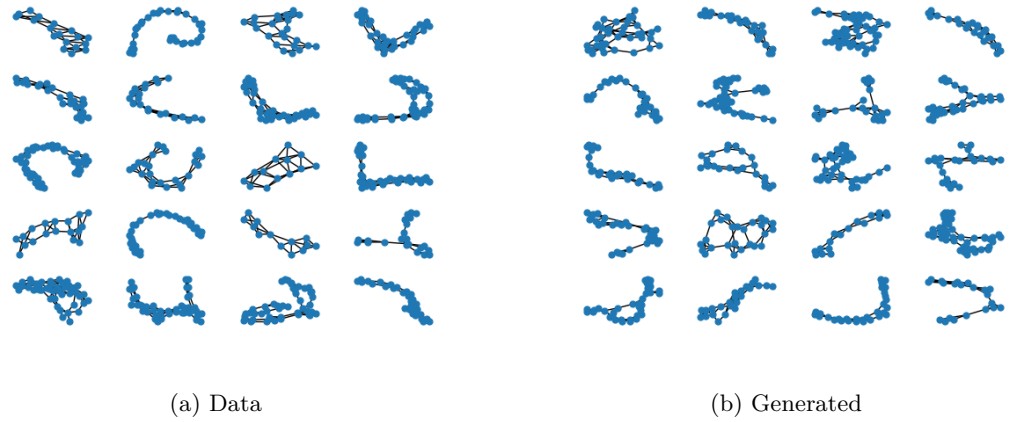

(a) Data

(b) Generated

Figure 8: Example of graphs from the Community-Small dataset and from generated samples.

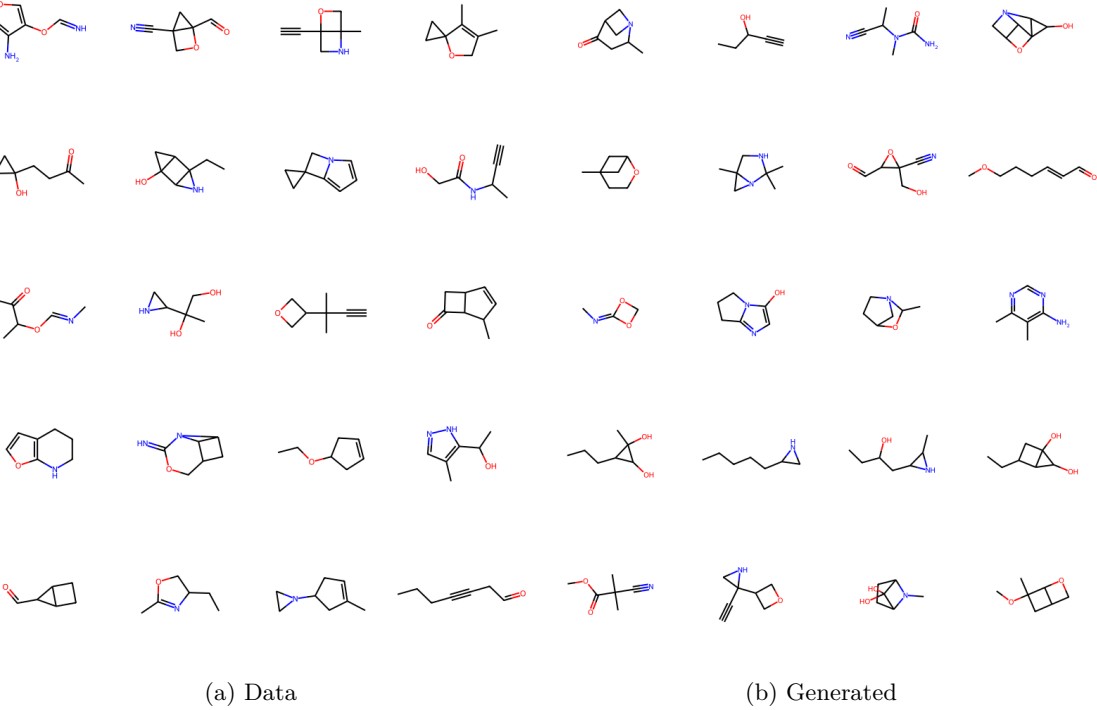

(a) Data

(b) Generated

Figure 9: Example of graphs from the Qm9 dataset and from generated samples.

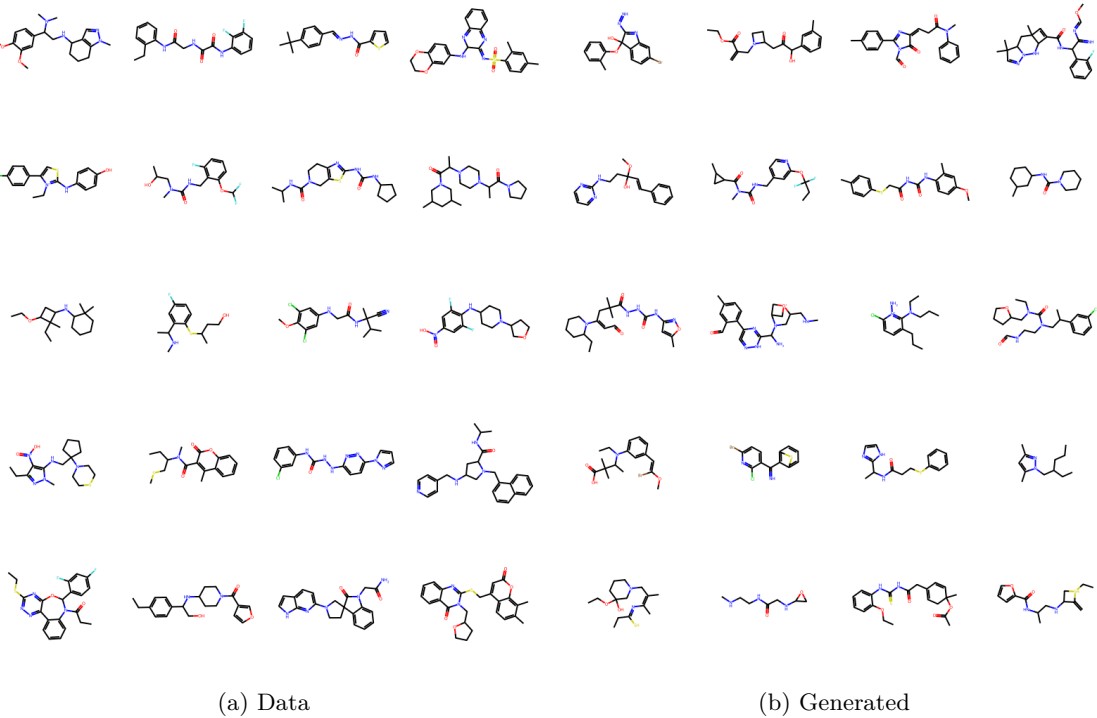

(a) Data                                                          (b) Generated

Figure 10: Example of graphs from the Zinc250K dataset and from generated samples.

## C.1  Additional experimental results

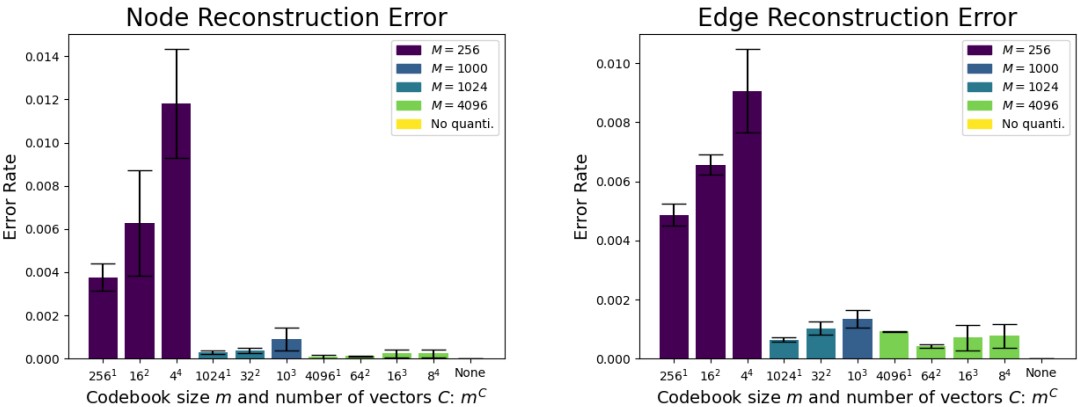

Figure 11: Effect of the codebook size and the partitioning on node (left) and edge (right) reconstruction errors. We report the best error rates averaged over 3 runs. The black lines indicate the standard deviations.

## D  Feature augmentation

We use 4 types of feature augmentations: $p$-path information, spectral information, cycles information, and random features.

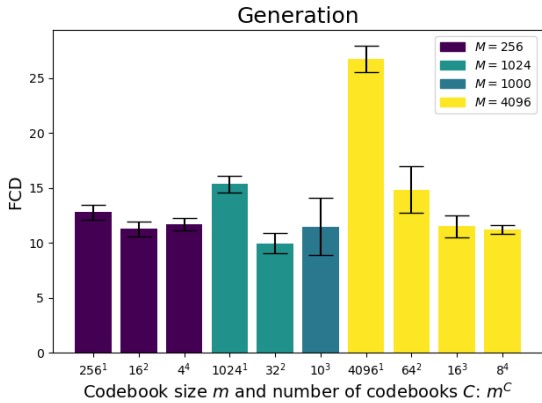

Figure 12: We report the average of the best reconstruction loss over 3 runs. The black lines indicate the standard deviations.

## D.1   Paths feature

The $p$-paths information is the number of paths of length $p$ connecting to nodes. We remind that path is a walk in which all edges and vertices are distinct. However, we allow the first and last vertices to be the same (cycles). We only compute the paths up to $p = 3$. We use the following formulas, where $A$ is the adjacency matrix, $D$ is the diagonal matrix of degrees and $I$ is the identity matrix. We assume that the original graph is connected.

$P_1 = A$

$P_2 = A^2 - D$

$P_3 = A^3 - AD - (D - I)A$

We incorporate this information as a vector $\boldsymbol{e}^p_{i,j}$ of edge attributes $\boldsymbol{e}^p_{i,j} = [p_{1(i,j)}, p_{2(i,j)}, p_{3(i,j)}]^T$. We adapt the definition of the neighborhood to include all nodes that are reachable with one of the paths. Similarly, we incorporate the $p$-degrees $p_{j,i}$ ($\boldsymbol{p}_j = P_j\mathbf{1}$) as a vector of node attributes.

## D.2   Spectral feature

The spectral features that we use are the $k$ eigenvector associated with the $k$ smallest eigenvalue of the Laplacian $L$, which is defined as $L = D - A$. Each value in the eigenvectors are associated with one node. By taking the $k$ first eigenvector, we create a vector of size $k$ of synthetic node attributed.

## D.3   Cycles feature

The $c$-cycle information consists of the number of cycle of size $c$ a node is part of. As in Digress Vignac et al. (2023), we incorporate the information of $c \in \{3, 4, 5\}$. The formulas for the computation are given in the Appendix of Vignac et al. (2023).

## D.4   Random feature

The random features are simply random value sampled from a known distribution. We used the standard Gaussian distribution and a vector size of 4.

## D.5   Examples of indistinguishable graph substructures

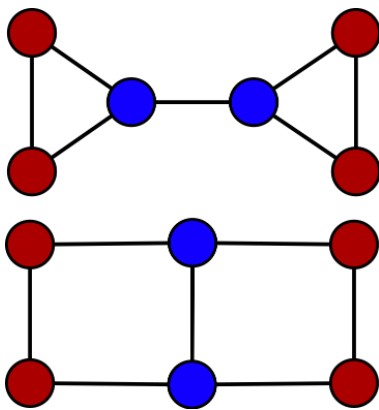

Figure 13: Assuming these two graphs are unannotated, any standard MPNN yields the same node features for all the red nodes as well as for all the blues nodes. It is an example, where MPNN cannot distinguish simple substructures as triangles and squares. However, any of the above methods yield synthetic features that theoretically allow MPNN to distinguish the nodes in the upper graph from the ones in lower graph.