# OpenReview forum: "Discrete Graph Auto-Encoder"
_TMLR — Accepted by TMLR_

### Review · Reviewer_xt8A · 2023-12-24

**Summary Of Contributions:**

In the submitted manuscript, the authors propose a model for graph generation. Their proposed DGAE model combines a graph autoencoder with a clustering step, a sorting step of the learned embeddings, and a Transformer model that is fit to the sorted embedding sequences. They present a range of experiments on five different datasets in which they demonstrate good performance of their proposed DGAE model.

**Audience:**

Yes

**Broader Impact Concerns:**

I do not have any concerns about this work that would necessitate a broader impact statement.

**Claims And Evidence:**

Yes

**Requested Changes:**

Generally speaking, it seems to me that many changes are required for the submitted manuscript to be ready for publication. It is my impression that the newly introduced ideas need to be presented more clearly and in particular your model needs to be better described. I now present my proposed changes in detail.


1] It would significantly strengthen Section 3.2 if you could discuss Graph Autoencoders [1] and related literature in greater detail. Given that your model is also a graph autoencoder, this would help place your model into context.

2] It would be better if you could describe your model in greater detail. I feel I may have misunderstood certain elements of your model despite my best efforts to grasp the sequence in which you apply the different model components. This would certainly help my understanding of its different components.

2.1] It is unclear to me how and for what reason the partitioning of the learned embeddings is performed. According to which criterion do you partition the learned representations and why do you see the need for partitioning? It would be great if you could describe this in greater detail in your paper.

2.2] I was unable to understand why you deem the clustering step in your architecture to be necessary. i) Without experimental evidence, I am not convinced that it removes "superfluous information". When clustering it seems to me that you may be losing valuable information. It would be great if you could clarify this part in the paper by stating the practical reason that necessitates the use of a clustering operation. ii) In particular, I wonder whether it would be possible to run our architecture without the clustering step. Such an experiment would help motivate your use of clustering. iii) It might furthermore be interesting to explore learnable alternatives to clustering such as a DiffPool layer [1].


2.3] I have several questions regarding your choice of decoder. i) Using an MPNN as a decoder in a graph autoencoder is not a standard choice. It is more common to work with activated inner products [2], i.e., \sigma(ZZ^T), as a decoder. It would be good if you could motivate your choice of decoder and potentially run an ablation experiment for it. ii) In the decoder it is unclear to me how you get back to having $n$ representations when it follows the clustering step. To me, it seems that the clustering step reduced the number of represented objects the number of centroids. iii) It is furthermore unclear to me why you chose to work with a complete graph instead of the original graph in the decoder. iv) If nodes are represented via their corresponding clustering centroids then they cannot be distinguished in a complete graph. Is this indeed the case?


2.4] In your feature augmentation scheme you draw virtual edges between nodes that are connected by paths of length up to $p.$ i) It would be interesting if you could discuss whether these edges are then considered in addition to the original edges in the GNN encoder and whether the virtual edges distinguish themselves from the originally present edges in any way. ii) Since the computational complexity of GNNs scales linearly in the number of edges it seems to me that your proposed feature augmentation scheme might add a significant computational expense to the model. It would be good if you could alleviate this concern by not only recording the infernce but also the training and potential pre-processing time to your experiments in Section 5. iii) While this is certainly not required for publication, it may be a nice addition to your paper to make a theoretical statement on what kind of graphs, that are indistinguishable using traditional MPNNs, can be distinguished as a result of your feature augmentation scheme.

2.5] In Section 4.3 you state that you sort node representations by index. i) It is unclear to me which indexes you use in your sorting. Are you sorting by first entries of the learned embeddings? ii) While you discuss that the clustering step in your architecture is not differentiable and how this issue is resolved in your training, you do not mention the fact that the sorting operation you perform is not differentiable either. It may be good to also discuss how you manage to backpropagate through the sorting operation.

2.6] In Section 4.3.2 you state that you "linearly project these vectors so that all the input vectors have the same dimensiond model". It is unclear to me why there were input vectors of unequal dimension at this stage. It might help explain your model more clearly if you could specify the source of these vectors and why there are vectors of unequal dimension.


3] In Section 5.3 it would also be good to mention the number of parameters of the different compared models. It seems to me that your model involves many parameters, which may result in an unfair comparison if fewer parameters were fitted for your baseline models.

4] It would be interesting if you could include the recent DiGress model [3] in your experiments to place your performance in the context of recent high-performing models.

5] Minor Changes:
-In the abstract the formulation "and there is no known fast algorithm." is a bit ambiguous and out of context. On first reading, it's unclear what kind of algorithm you are referring to here.

-It is unclear what kind of graph generator you draw your random graph from here "we refer to the random graph with G" and I do not see where these random graphs are used later on. So, I would either specify that you consider the Erdos-Renyi model, or a comparable one, or to remove this statement. Similarly, I do not understand what the definition of the probability of a graph in Section 2.2 adds to your paper. I do not see it being used later on and it could confuse certain readers.

- The statement "algorithms that uniquely sort any graph" is imprecise. I think you are referring to algorithms that uniquely sort nodes in a graph.

- It would be nice to add a few references to Section 2.3.1 to more firmly root your statements in the literature. In general, would it be nice to point to specific papers using BFS in graph generation early on in your introduction to ensure that readers become aware that this was indeed an approach that was proposed and successfully used.

- The following description of GNNs could easily be misunderstood "Since the cardinality of the input and the output of such function must match, GNNs are said to be ’flat’." While indeed the number of learned representations corresponds to the number of nodes throughout the architecture (for the majority of GNNs), the dimension of each learned representation often varies at the different layers.

- When you say that "Our solution is related, but not identical, to the recent work presented in Feng et al. (2022)." it would be nice to state specific differences between your approaches to make your statement more readily credible to the reader.

- In Equation (10) you have the notation $z_{<i,1}.$ I am unfamiliar with the notation $<i$ in your subscript. Could you please define this?

- The formulation "we introduce a virtual node (z 0,1 , ..., z 0,C ) made of vectors of zeros." is unclear to me. How can a single node be "made of" several vectors?

- It would be good if you could more clearly define what "orbits with four nodes" are.

- In Section 5.3.1 you state that your model "remains competitive" in the generation of valid molecules, while it performs almost 20% worse on the Zinc dataset. It seems to me that your discussion of these results should be weakened to be more realistic.

- In Figure 3 the legend is rather small on a printed copy and the colour scheme makes it challenging to differentiate the different lines.



[1] Ying, Zhitao, et al. "Hierarchical graph representation learning with differentiable pooling." Advances in neural information processing systems 31, 2018.

[2] Kipf, Thomas N., and Max Welling. "Variational graph auto-encoders." . NeurIPS Workshop on Bayesian Deep Learning, 2016.

[3] Vignac, Clement, et al. "Digress: Discrete denoising diffusion for graph generation." ICLR, 2023.

**Strengths And Weaknesses:**

**Strength**

1] The way in which you discuss related work in Section 3 is rather nice.

2] In general the mathematical notation is often well-defined and clear.

3] In Section 5.1.2 you provide a good discussion of the shortcomings of different metrics used for graph generation on molecular datasets and propose a seemingly improved evaluation scheme.

**Weaknesses**

1] In the abstract and introduction you heavily focus the discussion on the graph isomorphism problem. I am not convinced that this is the best aspect of your model to place the focus on. The fact that GNNs are permutation equi-/invariant is well-known and the resulting advantages have also been discussed in many papers. I therefore, find your initial discussion of the two possible approaches to modelling graphs (BFS and GNNs) not to be overly insightful. I suggest you shift the focus from graph isomorphism initially.

2] I am not convinced of the necessity of the clustering step in your architecture. You make several comments speaking to its advantages such as,

- "To ease the modeling of the latent distribution, we remove unnecessary information and reduce the size of the latent space by clustering the node representations." and equally, "The primary objective of clustering is to ease the learning of the latent distribution. We achieve this thanks to two main consequences of clustering: first, by removing superfluous information". It is unclear to me whether you truly remove "unnecessary information" and I don't see evidence in your experiment substantiating this statement.

- "clustering also ensures better coverage of the latent space". I don't see how reducing the number of utilised embeddings from a set of embeddings containing one embedding per node to a set of embeddings containing one embedding per centroid ensures better coverage of the latent space.

3] It seems to me that your model should be presented in greater detail and more clearly. (see Point 2 in the Requested Changes for more detail.)

---

> ### Author Response · Authors · 2024-01-25
> **Answer**
>
> We thank you for the quality and precision of your feedback.
>
> Regarding the points in ‘weaknesses’:
>
> 1. Graph representation: We agree with you on the fact the graph isomorphism problem is well understood and that we do not make significant (if any) contribution regarding the understanding of the problem. However, these elements are what make the specificity of graph generative model and they explain the choice we are doing for our model. So we are still convinced that they are necessary. However, we rewrote this abstract to put emphasis on the contributions of our model.
>
> 2. Clustering: We first notice that the description of our model lacked clarity and was subject to misunderstandings. In particular, the use of the term ‘clustering’ is very misleading in the context of graph, since node clustering within a graph is a well-established task in graph theory. In the updated version, we prefer the term quantization, which refer to a whole family of methods much closer to what we are actually doing in our model.
> In addition, some unprecise formulation such as ‘we remove unnecessary information’ (where we meant to say that we learn to keep only the most valuable information for reconstruction) have been removed.
> We have deeply reformulated the model description, improving the presentation of our motivations for quantization (clustering). We hope that it answered your concerns. Please, let us know if you still have concerns regarding the quantization/clustering step.
>
> 3. We already answered this point in the general comments. We revised the text, including a complete rewriting of the introduction of Section 4 and section 4.1.3 (revised version). More details on the changes are given hereunder.
> Detailed responses to the ‘requested changes’ following your numeration :
>
>     1. 1. We have added a paragraph about Graph Auto-Encoders, even though there are not generative for graphs there are, indeed, related to our work.
>
>     2. We significantly changed the model description. In Section 4, we have re-written completely the introduction and the Section 4.1.3.
>
>         2.1. Partitioning is essentially done for practical reasons. It prevents of having to model a discrete distribution with a very large number of categories during the second step. I think that this aspect is much clearer in the new version.
>
>         2.2. I think, your comment 2.2 regarding the clustering comes from a misunderstanding of what we meant by clustering (again using ‘clustering’ was confusing). As said in the introduction, we have replaced the term clustering by quantization, which should be clearer. The new version of Section 4.1.3 should be also clearer, with that respect.
>
>             2.2.1. Indeed, there is no superfluous information. We meant to say that we remove the less valuable information for reconstruction. We removed this formulation in the new version.
>
>             2.2.2. As mentioned in the general answer, we can run the architecture of the auto-encoder without quantization (first step), but then our auto-regressive model cannot learn the latent representation in the second step.
>
>             2.2.3. We think that Diffpool, doing graph pooling, is irrelevant (again misunderstandings related the term clustering).
>
> to continue...

---

> ### Author Response · Authors · 2024-01-25
> **Answer**
>
> 2.3.
>             2.3.1. Using an activated inner products [2], i.e., \sigma(ZZ^T), as a decoder has at least 2 drawbacks. First, it can only output a scalar. Therefore, it is not suited to predict annotated edges. Second, we assume that a same pair of latent node representation may not results in the same edge prediction depending on the interaction with the other node embeddings, but the activated inner products decoder can not take these interactions into consideration.
> Moreover, it seems that the inner product decoder is only used in [2] which is not a generative model. It is not enough to make of this decoder a ‘standard’ choice.
> That being said, if you think that it is still useful and interesting, we can run an ablation with such a decoder on one of the experiment with unannotated graph.
>
>             2.3.2. We always have n representations. We make it very explicit in the revised version.
>
>             2.3.3. The original graph is what we try to model. So, we do not have access to it at generation time. Consequently, our decoder cannot use this information. The goal of our decoder is to recover the graph structure by predicting the absence or presence of edge between all pairs of node. We note also that this structure is standard and used by all permutation-equivariant methods for graph generation.
>
>             2.3.4. Indeed all the nodes in the same graph (except those belonging to the same orbit, i.e. having the same structural role), should have different latent representations. Given the high level of reconstruction accuracy, we assume that they are indeed different.
>
>         2.4.
>             2.4.1. In the dedicated Section, we slightly reformulate this part. It should be clearer now. The paragraph in the revised version:
> ‘we first create virtual edges (with a vector of zeros as attributes) between nodes that are connected by paths of length 2 to p (the edge between adjacent nodes, i.e. connected by a path of length 1, are already represented by a vector of edge attributes). We then concatenate a synthetic p−dimensional vector to each edge attribute, whose ith entry corresponds to the number of paths of length i connecting the two endpoints.’
>
>             2.4.2. Indeed, our edge feature augmentation scheme increases the computational complexity of our model. However, it does not affect drastically either the training time nor the preprocessing time. Generation is not affected, since we do not use the encoder for generation. In addition, since MPNN uses the same set of parameter for all the edges, it barely increases the number of model parameters.
> We added a new table in our experiment, showing training and preprocessing time with and without our feature augmentation method.
>
>             2.4.3. We totally agree that is would be a nice additional feature. Unfortunately, it is unrealistic for us in terms of load of work. It would be a full paper on its own.
>
>         2.5.
>             2.5.1. We are talking about the indices corresponding to the codewords. In the revised version, we made explicit the link between the codewords and the indices and, in Section 4.2.1 we refer to section 4.1.3 where the sequence of indices are defined.
>
>             2.5.2. We perform the sorting operation as a preprocessing operation of our second step. Therefore, we do not need to compute any gradient through the sorting operation.  We add a paragraph to explicit this point.
>
>         2.6. As explain in this paragraph, we concatenate the known partitions to create the input vector. Since, the number of known partitions vary so does the size of the input vector. We reformulated the manuscript to make this clearer.
>
>     3. We do not think that the number of parameters is a limiting factor in graph generation. The number of parameter of graph generative model are generally way below the models used in other deep learning domain as computer vision or NLP. Neverthless, we added this information in the appendix.
> Unfortunately, the comparison with other models do not seems possible. The number of parameters changes depending on some hyperparameters (in particular the number of layers, blocks and/or layer units are missing) changing across experiments. In most of the baselines papers this information is missing.
>
>     4. We added DiGress and GraphARM in our experiment baselines.
>
>     5. We revised our paper following your recommendations. Regarding the question in 7th bullet point: yes, we represent a single node with several vectors, as it should be clear in the revised version.

---

### Review · Reviewer_2GRG · 2024-01-02

**Summary Of Contributions:**

The paper introduces the **Discrete Graph Auto-Encoder (DGAE)**, a novel approach that addresses challenges related to the equivariance property in graph generative models. DGAE employs two key strategies to tackle these issues. First, it projects a graph into a set using a discrete and invariant encoder, thereby eliminating the need for costly graph matching procedures.

To enhance the latent space, DGAE incorporates a clustering procedure, facilitating the learning of informative latent embeddings. This is achieved through a codebook containing various codeword sequences. Notably, the use of sorted sets enables the autoregressive learning of unique representations.

The decoder in DGAE incorporates a specialized **2D attention mechanism** designed for modeling sequences in two dimensions. Additionally, the paper introduces various feature augmentation techniques and empirically validates these strategies through multiple experiments on diverse datasets. The results are compared against various baselines, providing a comprehensive evaluation of the proposed model.

**Audience:**

Yes

**Broader Impact Concerns:**

No particular need for a Broader Impact Statement,

**Claims And Evidence:**

Yes

**Requested Changes:**

The paper is well-written, addressing my questions and concerns about the model's training and technical aspects through multiple experiments. I recommend the authors to explore recent graph diffusion generative models for comparison with the current state-of-the-art. Additionally, it would be beneficial for the authors to provide a more detailed discussion and ideas on how to scale their proposed model to larger graphs. Overall, the paper is very good.

**Strengths And Weaknesses:**

## Main Strengths

1. The paper combines different but prominent strategies tackling the equivariance issues raised when working with graph structures, with impressive performance.

2. Introduces a novel 2D transformer architecture modeling 2D sequences.

3. It offers a new technique for graph augmentation, named p-path and importantly validates graph augmentation techniques in an ablation study showcasing the importance of such procedures.

4. It offers competitive performance against prominent baselines based on multiple networks while massively outperforming the baselines in the generation time.

## Weaker Elements

1. The paper does not contain any more recent graph diffusion generative models but rather an older model from 2019. They could potentially introduce more recent works such as Digress ([link](https://openreview.net/forum?id=UaAD-Nu86WX)).

2. The defined model produces a two-stage training which, as also mentioned by the authors, can increase the training complexity, as it defines multiple hyperparameters.

3. It does not offer any concrete ideas on how to potentially scale the model to large graphs, a very important aspect.

---

> ### Author Response · Authors · 2024-01-25
> **Answer**
>
> We thank you again for the careful reading and the comments of our paper.
> Regarding the weaker elements. We have included DiGress as baseline (see general remarks).
> Regarding the number of hyper-parameters associated with the two stage modeling and the scalability, we are aware of these limitations and discuss them in the conclusion. Our paper is already long, so these issues are doing to be addressed in further works.

---

### Review · Reviewer_uXMb · 2024-01-18

**Summary Of Contributions:**

This paper introduces a new method for the generative modeling of graphs.

This is achieved with an autoencoder model with enforced clustering in the latent space to produce representations of the graph, and an auto regressive model with a new transformer architecture that models the latent distribution as the generative component of the model.

The method is compared on a number of tasks with reasonably convincing results, alongside ablation studies of the method.

**Audience:**

Yes

**Claims And Evidence:**

Yes

**Requested Changes:**

I would like to see a better discussion of related work in the experiments section, comparing to more recent works, and state of the art methods that are of a different type to this work. There are particularly some related diffusion works not discussed.

One ablation not included that I think should be is not using a codebook at all. It would be very interesting to see if this feature actually helps, as it is not evident to me that it will.

It is not clear to me that the autoregressive model will generate sorted lists of codebook keys - could the authors comment on this?

p 13. seciton 4.4 paragraph 2, states "there are 4 reasons..." but only 3 are listed as "secondly" is skipped.

**Strengths And Weaknesses:**

Overall the paper is well written and easy to understand. The description of the problem, state of the art, and solution is clear and choices are well motivated throughout the text.

The experiments are convincing enough that this method works, although generic graph generation is not my area of expertise enough to comment on the choice of baselines in the simple graphs section. I was surprised not to see any methods from 2023 compared in the paper - given the pace of the field I would usually expect something new to have occurred in the last year that is of relevance.

In the experiments on molecular graphs, perhaps some additional references could be included. For example:
- Equivariant Diffusion for Molecule Generation in 3D
- Geometric Latent Diffusion Models for 3D Molecule Generation
are both methods that perform well on the QM9 task, one of them performing better than the method in this paper in fact.

The ablation studies in the end of the appendix are good to see and clearly demonstrate benefits of the different components of the model

---

> ### Author Response · Authors · 2024-01-25
> **Answer**
>
> We thank you again for the careful reading and the comments of our paper.
> We discuss two of your concerns in the general comments. So, we focus here on the other ones here: the inclusion of further molecular graph, namely ‘Equivariant Diffusion for Molecule Generation in 3D’ and ‘Geometric Latent Diffusion Models for 3D Molecule Generation’ and the question related to the auto-regresssivity.
> We agree on the fact to the two models are recent and very powerful model for molecule generation.  However, the comparison with these model are complicated for several reasons.
> First, they are not graph generative models, in the sense that they model molecules as 3D point cloud (inferring the molecule bond as a post-hoc prediction step).
> Second, they use additional domain-knowledge information, in particular the atom positions, making comparison unfair. Third, QM9 is the only common dataset between the two approaches (since this approach needs the atom position they cannot train on Zinc250, which does not contain this information). Finally, because of this differences, they also use different metrics. Validity and uniqueness , metrics that have limitations as discussed in our paper, are the only shared metrics between the two approaches.
> We included a paragraph discussing this and reporting validity of these models.
> About why the auto-regressive model will generate sorted indices, there are two reasons. First, the training instances are sorted, so that if the auto-regressive model learns the sequence correctly it should learn the ordering as it is in the sorted training set. Second, we enforce the model to yield sorted indices by masking the indices lower than the last generated index. We have reformulate the last paragraph of Section 4.2.2 to clarify that.
>
> Regarding p 13. section 4.4 paragraph 2, we thank you for pointing out this typo and did the correction.

---

### Author Response · Authors · 2024-01-25
**Answer to reviewers**

First and foremost, we wish to extend our deepest gratitude for your meticulous review of our article and for your constructive feedback. Initially, we will address the common concerns raised, followed by a detailed response to each of your specific comments.
We have identified three main points of improvement regarding our paper: 1) including more recent baselines, 2) improving the clarity in the presentation of our model 3) running an additional ablation study about quantization.

1)
A major issue, you all have listed, concerned the lack of most recent baselines. As suggested, we have included DiGress in the revised version. We also included GraphARM as baseline model (GraphARM, icml 2023).  We also want to add these models on the generation time experiment. However, we need more time to run these experiments. We will comeback to you about this.

2)
Reviewer xt8A pointed out the lack of clarity regarding the presentation of our model. We agree that some elements of our presentation could be misleading. In particular, in this context, the term ‘clustering’ is too much associated with community detections and node clustering within a graph, which are not what we are doing. So, in the revised version, we prefer the term ‘quantization’ (The concept is actually correct if we consider that the latent space is continuous, even though the support of latent distribution is sometimes discrete).
In the new version, we rewrote completelty the introduction of Section 4 (which includes now the content of the previous Section 4.1) and the Subsection 4.1.3 (4.2.3 in the previous version) to improve the clarity of our model presentation. We also brought additional explanation to all points mentioned.
We really hope that the new version is easier to follow. If some points remain unclear after these modifications, please let us know.  We are willing to improve the presentation further.

3)
Two of the reviewers, xt8A and uXMb, questioned the utility of the clustering/quantization step and asked for an ablation study on this points. As clarified in the new version, we indeed lose information during this step, and in fact the reconstruction performances decreases when the quantization is coarser (smaller codebook), but this is not true for generation (where model with larger codebook size do not perform better).
However, quantization is necessary to train the second step of our model. By consequent, we can add an ‘no clustering/quantization’ for the evaluation of the auto-encoder, but not for generation.
In addition to these comments, we include another significative change to answer a comments from xt8A. We present 2 new empirical results one on the dictionary usage, reporting the dictionary usage for various partition and codebook sizes and another on the effect of our feature augmentation method on the preprocessing and training time.
Again, we warmly thank you for your feedback. We are convinced that the changes based on your reviews significantly improve the overall quality of our paper.
Due to characters restriction, we answer more precisely to each of your remarks and questions independently.

---

### Author Response · Authors · 2024-01-30
**Small modifications**

Dear reviewers,

As announced in the previous comment, we have now added the generation time for DiGress. Unfortunately, we did not find any public repository for GraphARM and our attempt to contact their author is so far unsuccessful.
In addition, we also made some little formal adjustments to the supplementary material.
We thank you again for your time and attention.

---

### Author Response · Authors · 2024-02-07
**Small correction**

Dear reviewers,
We just noticed that, in the last revised manuscript, we let appear a sentence that we should have deleted  (Section 4.2.1 4th paragraph).
We just edited the text to removed it. We apologies for this late change.

---

### Decision · Action_Editor_hkyd · 2024-02-29

**Recommendation:** Accept with minor revision

**Comment:**

The paper was reviewed by three reviewers. The reviewers raised concerns about the significance of the empirical results since the proposed model was not compared against any recent method. The reviewers also complained about the absence of an ablation study to investigate whether the clustering/quantization component actually helps. One reviewer also expressed concerns about the clarity of presentation of the approach and requested several changes. Most of the reviewers' concerns were addressed in the revision, and all three reviewers are in favor of accepting this paper. Two reviewers recommended "Leaning Accept", and one reviewer recommended "Accept". I am recommending accept with a minor revision, and I request that the claim discussed above (that the proposed model outperforms state-of-the-art models ) is modified in the final version of the manuscript. Also, while reading the paper, I noticed that the authors use \cite instead of \citep throughout the manuscript and that there are a few typos, e.g., page 1: "and mitigate their" -> "and mitigates their". I expect the authors to fix those presentation issues in the final version.

**Audience:**

Graph generative models have attracted a lot of attention recently within the graph learning community. Therefore, the topic and findings of the paper are of interest to some individuals in TMLR's audience.

**Claims And Evidence:**

Overall, most of the claims made in the paper are well supported by accurate evidence. In page 2, the authors claim that the proposed model "outperforms existing state-of-the-art models across multiple metrics and datasets". This claim is not supported by the reported experimental results.